# Refined Convergence and Topology Learning for Decentralized Optimization with Heterogeneous Data

**Batiste Le Bars**
Univ. Lille, Inria, CNRS, Centrale Lille,
UMR 9189, CRIStAL, F-59000 Lille
`batiste.le-bars@inria.fr`

**Aurélien Bellet**
Univ. Lille, Inria, CNRS, Centrale Lille,
UMR 9189, CRIStAL, F-59000 Lille
`aurelien.bellet@inria.fr`

**Marc Tommasi**
Univ. Lille, Inria, CNRS, Centrale Lille,
UMR 9189, CRIStAL, F-59000 Lille
`marc.tommasi@inria.fr`

**Erick Lavoie**
EPFL, Lausanne, Switzerland
`erick.lavoie@epfl.ch`

**Anne-Marie Kermarrec**
EPFL, Lausanne, Switzerland
`anne-marie.kermarrec@epfl.ch`

## Abstract

One of the key challenges in decentralized and federated learning is to design algorithms that efficiently deal with highly heterogeneous data distributions across agents. In this paper, we revisit the analysis of Decentralized Stochastic Gradient Descent algorithm (D-SGD) under data heterogeneity. We first exhibit the key role played by a new quantity, called *neighborhood heterogeneity*, on the convergence rate of D-SGD. Neighborhood heterogeneity provides a natural criterion to learn data-dependent and sparse topologies that reduce the detrimental effect of data heterogeneity on the convergence of D-SGD. For the important case of classification with label skew, we formulate the problem of learning a topology as a tractable optimization problem that we solve with a Frank-Wolfe algorithm. As illustrated over a set of experiments, the learned sparse topology is showed to balance the convergence speed and the per-iteration communication costs of D-SGD.

## 1 Introduction

Decentralized and federated learning methods allow training from data stored locally by several agents without exchanging raw data. One of the key challenges is to deal with data heterogeneity: as each agent collects its own data, local datasets typically exhibit different distributions. In this work, we study this challenge in the context of fully decentralized learning algorithms. Those algorithms, such as the celebrated Decentralized SGD (D-SGD) [22, 23, 17, 35, 19], operate on a graph representing the communication topology, i.e. which pairs of nodes exchange information with each other. Such topology rules a trade-off between the convergence rate and the per-iteration communication complexity of the algorithms [34]. Choosing a good topology is therefore an important question, but remains a largely open problem in the presence of data heterogeneity.

In this work, we fill the theoretical gap that currently exists on this question. Our first contribution is a refined convergence analysis of D-SGD which introduces a new quantity, called *neighborhood heterogeneity*, coupling the topology with the data distributions. Our results exhibit sharper rates than

Workshop on Federated Learning: Recent Advances and New Challenges, in Conjunction with NeurIPS 2022 (FL-NeurIPS'22). This workshop does not have official proceedings and this paper is non-archival.

those of the state-of-the-art [17] and demonstrate that the impact of the topology on the convergence rate of D-SGD does not only depend on its connectivity: it also depends on its capacity to compensate the heterogeneity of local data distributions at the neighborhood level.

Our second contribution deals with the problem of learning a good *data-dependent* topology. We argue that neighborhood heterogeneity provides a natural objective and show that it can be effectively optimized in practice in the important case of classification with label distribution heterogeneity across nodes [16, 13, 2]. We solve the resulting problem using a Frank-Wolfe algorithm [11, 15], allowing us to track the quality of the learned topology as new edges are added in a greedy manner. To the best of our knowledge, our work is the first to learn the graph topology for decentralized learning in a way that is data-dependent, controls communication costs, and is theoretically justified.

## 1.1 Related Work

**Algorithmic improvements to decentralized SGD.** Significant work has been devoted to extensions of D-SGD. We can mention those based on momentum [1, 12, 24, 38], cross-gradient aggregations [10], gradient tracking [18] and bias correction (or variance reduction) [31, 37, 36, 14]. Many of these schemes are able to reduce the order of the term that depends on the heterogeneity but remains impacted by strong heterogeneous scenarios. We stress that the above line of research is complementary to ours as it is based on modifications of the D-SGD algorithm (which often requires additional computation and/or communication). In contrast, our work does not modify the algorithm: we provide a refined analysis and a method to learn the topology. We believe that our results can be combined with the above algorithmic improvements, but leave such extensions for future work.

**Choosing and learning good topologies for decentralized learning.** There is a long line of research on choosing (e.g. grid, circle, exponential graphs) [6, 28, 35] or learning [3, 34, 26, 35] topologies maximizing their spectral gap. Unlike our approach, these methods simply seek to increase the connectivity of the topology, but they do not take into account the data distributions across nodes.

Until now, [2] was the only work leveraging the distribution of data in the design of the topology. Focusing on classification under label skew, they propose a heuristic approach that consists of interconnected cliques, where the proportion of classes in each clique should be as close as possible to the global proportion. Our approach is more flexible as it can learn more general topologies, and provides full control over the sparsity of the topology. Furthermore, our topology learning criteria is theoretically justified, while the one in [2] is only supported by empirical experiments. We think however that the ideas of the present paper could pave the way for a theoretical analysis of their work.

## 2 Preliminaries

In decentralized and federated learning, a set of $n \in \mathbb{N}^\star$ agents (nodes) with their own data distribution collaborate in order to learn a single parameter $\theta \in \mathbb{R}^d$ optimizing the global consensus objective [22]:

$$f^* \triangleq \min_{\theta \in \mathbb{R}^d} \left[ f(\theta) \triangleq \frac{1}{n} \sum_{i=1}^{n} f_i(\theta) \right], \tag{1}$$

where each $f_i(\theta) \triangleq \mathbb{E}_{Z_i \sim \mathcal{D}_i}[F_i(\theta; Z_i)]$ is the local objective function associated to node $i$. The random vector $Z_i$ is drawn from the data distribution $\mathcal{D}_i$ of agent $i$, having support over a space $\Omega_i$, and $F_i$ is its *pointwise* loss function (differentiable in its first argument). Note that the distributions $\mathcal{D}_i$ can differ much, leading the local optima $\theta_i^\star \triangleq \arg\min_\theta f_i(\theta)$ to be far from the global optimum $\theta^\star$ of (1).

To collaboratively solve (1) in a fully decentralized manner, the agents communicate with each other over a directed graph. The graph topology is represented by a matrix $W \in [0, 1]^{n \times n}$, where $W_{ij} > 0$ means that agent $i$ exchange messages with agent $j$, while $W_{ij} = 0$ (no edge) means that node $i$ and $j$ do not communicate with each other. The choice of topology $W$ affects the trade-off between the convergence rate of the algorithms and the communication costs. Indeed, more edges imply higher communication costs but often faster convergence. Communication costs, or *per-iteration complexity*, are often regarded as proportional to the maximum (in or out)-degrees of nodes in the topology, representing the maximum (incoming or outcoming) load of a node [22]:

$$d_{\max}^{\text{in}}(W) = \max_i \sum_{j=1}^{n} \mathbb{I}[W_{ji} > 0], \quad d_{\max}^{\text{out}}(W) = \max_i \sum_{j=1}^{n} \mathbb{I}[W_{ij} > 0]. \tag{2}$$

D-SGD [22, 17] is a popular fully decentralized algorithm for solving problems of the form (1). It requires that $W$ is a *mixing* matrix i.e. a doubly stochastic matrix, and in the rest of the paper, we

will use the terms topology and mixing matrix interchangeably. D-SGD is recalled in Algorithm 1 of Appendix A.1. At iteration $t$, each node $i$ first updates its local estimate $\theta_i^{(t)}$ based on $\nabla F_i(\theta_i^{(t)}, Z_i^{(t)})$, the stochastic gradient of $F_i$ evaluated at $\theta_i^{(t)}$ with $Z_i^{(t)}$ sampled from $\mathcal{D}_i$. Then, each node aggregates its current parameter value with its neighbors according to the mixing matrix $W^{(t)}$.

Recall some standard assumptions extensively considered in decentralized learning [4, 30, 22, 21, 19]:

**Assumption 1.** (*L-smoothness*) *There exists a constant $L > 0$ such that for any $Z \in \Omega_i$, $\theta, \tilde{\theta} \in \mathbb{R}^d$ we have $\|\nabla F_i(\theta, Z) - \nabla F_i(\tilde{\theta}, Z)\| \leq L\|\theta - \tilde{\theta}\|$.*

**Assumption 2.** (Bounded variance) *For any node $i \in [\![1, \ldots, n]\!]$, there exists a constant $\sigma_i^2 > 0$ such that for any $\theta \in \mathbb{R}^d$, we have $\mathbb{E}_{Z \sim \mathcal{D}_i}\big[\|\nabla F_i(\theta, Z) - \nabla f_i(\theta)\|_2^2\big] \leq \sigma_i^2$.*

**Assumption 3.** (Mixing parameter) *There exists a mixing parameter $p \in [0, 1]$ such that for any matrix $M \in \mathbb{R}^{d \times n}$, we have $\|MW^\mathsf{T} - \overline{M}\|_F^2 \leq (1-p)\|M - \overline{M}\|_F^2$, where $\|\cdot\|_F$ denotes the Frobenius norm and $\overline{M} = M \cdot \frac{1}{n}\mathbf{1}\mathbf{1}^\mathsf{T}$ is the column-wise average of $M$ repeated $n$ times.*

## 3 Joint Effect of Topology and Data Heterogeneity

In this section, we introduce a new quantity called *neighborhood heterogeneity* and we derive new convergence rates for D-SGD that depend on this quantity. These rates have several nice properties: (i) they hold under weaker assumptions than previous work (unbounded heterogeneity), (ii) they highlight the interplay between the topology and the data distribution across nodes, and (iii) they provide a criterion for choosing topologies not only based on their mixing properties but also based on data.

### 3.1 Neighborhood Heterogeneity

Given a mixing matrix $W$, our notion of neighborhood heterogeneity measures the expected distance between *the aggregated gradients in the neighborhood of a node* (as weighted by $W$) and *the global average of gradients*. In the rest of our analysis, we assume this distance to be bounded.

**Assumption 4** (Bounded neighborhood heterogeneity). *There exists a constant $\bar{\tau}^2 > 0$ such that:*

$$H \triangleq \frac{1}{n}\sum_{i=1}^{n} \mathbb{E}\Big\|\sum_{j=1}^{n} W_{ij}\nabla F_j(\theta, Z_j) - \frac{1}{n}\sum_{j=1}^{n}\nabla F_j(\theta, Z_j)\Big\|_2^2 \leq \bar{\tau}^2, \quad \forall \theta \in \mathbb{R}^d. \tag{3}$$

*where the expectation is taken with respect to $(Z_1, \ldots, Z_n) \sim \mathcal{D}_1 \otimes \ldots \otimes \mathcal{D}_n$.*

To better understand Assumption 4, let us decompose the neighborhood heterogeneity $H$ using a bias-variance decomposition, which leads to the following bound:

$$H \leq \frac{1}{n}\sum_{i=1}^{n}\Big\|\sum_{j=1}^{n} W_{ij}\nabla f_j(\theta) - \nabla f(\theta)\Big\|_2^2 + \frac{\sigma_{\max}^2}{n}\|W - \frac{1}{n}\mathbf{1}\mathbf{1}^\mathsf{T}\|_F^2, \text{ with } \sigma_{\max}^2 = \max_i \sigma_i^2. \tag{4}$$

This upper bound contains two terms. The first one is a *bias term*, related to the heterogeneity of the problem. It essentially measures how the gradients of local objectives differ from the gradient of the global objective when they are aggregated at the neighborhood level of the topology through $W$. The second one is a *variance term* closely related to the mixing parameter $p$ of Assumption 3: it is upper bounded by $\sigma_{\max}^2(1-p)$ and lower bounded by $\sigma_{\max}^2(1-p)/n$, see Proposition 3 in Appendix D.

In our analysis, we will use Assumption 4 in replacement of the *bounded local heterogeneity* condition used in previous literature [22, 23, 31, 1, 34, 17, 35]. We recall it below.

**Assumption 5** (Bounded local heterogeneity). *There exists a constant $\bar{\zeta}^2 > 0$ such that $\frac{1}{n}\sum_{i=1}^{n}\|\nabla f_i(\theta) - \nabla f(\theta)\|_2^2 \leq \bar{\zeta}^2, \forall \theta \in \mathbb{R}^d$.*

Assumption 5 has the same form as the bias term of $H$ in (4) but considers $W = I$ (i.e., ignoring the links between nodes). It requires that the local gradients should not be too far from the global gradient: the more heterogeneous the nodes' distribution, the bigger $\bar{\zeta}^2$. In contrast, neighborhood heterogeneity takes into account the mixture of gradients in the neighborhoods defined by $W$. Crucially, our set of assumptions (Assumptions 2-4) is less restrictive than those of previous work (Assumptions 2, 3, 5). To see this, we first show that our set of assumptions is implied by the latter (proof in Appendix D).

**Proposition 1.** *Let Assumptions 2-3 and 5 to be verified. Then Assumption 4 is satisfied with* $\bar{\tau}^2 = (1 - p)\left(\bar{\zeta}^2 + \bar{\sigma}^2\right)$, *where* $\bar{\sigma}^2 \triangleq \frac{1}{n}\sum_i \sigma_i^2$.

We can show that our set of assumptions (2-4) is strictly more general than Assumptions 2, 3, 5 by identifying situations where the bias-variance decomposition (4) is bounded while Assumption 5 is not verified. A trivial example is with the complete graph $W = \frac{1}{n}\mathbf{1}\mathbf{1}^\mathsf{T}$, for which $H = 0$ and thus $\bar{\tau}^2 = 0$, regardless of heterogeneity. More interestingly, some combinations of *sparse* topologies and data distributions can ensure that $\bar{\tau}^2$ remains small while $\bar{\zeta}^2$ can be arbitrary large. We give such an example in Appendix B. In Section 4, we will show how we can learn a *sparse* topology $W$ that (approximately) minimizes the neighborhood heterogeneity $H$.

## 3.2 Convergence Analysis

We now present the main theoretical result of this section: two new non-asymptotic convergence results for D-SGD under Assumption 4. The proof of this theorem is given in Appendix C.

**Theorem 1.** *Consider the D-SGD algorithm with mixing matrices* $W^{(0)}, \ldots, W^{(T-1)}$ *satisfying Assumptions 3 and 4. Assume further that Assumptions 1-2 are respected, and denote* $\bar{\theta}^{(t)} \triangleq \frac{1}{n}\sum_{i=1}^n \theta_i^{(t)}$. *For any target accuracy* $\varepsilon > 0$, *there exists a constant stepsize* $\eta \leq \frac{p}{8L}$ *such that:*

*Convex case:*

$$\frac{1}{T+1}\sum_{t=0}^T \mathbb{E}(f(\bar{\theta}^{(t)}) - f^\star) \leq \varepsilon \text{ as soon as}$$

$$T \geq \mathcal{O}\left(\frac{\bar{\sigma}^2}{n\varepsilon^2} + \frac{\sqrt{L}\bar{\tau}}{p\varepsilon^{\frac{3}{2}}} + \frac{L}{p\varepsilon}\right)r_0\,, \quad (5)$$

*Non-convex case:*

$$\frac{1}{T+1}\sum_{t=0}^T \mathbb{E}\|\nabla f(\bar{\theta}^{(t)})\|_2^2 \leq \varepsilon \text{ as soon as}$$

$$T \geq \mathcal{O}\left(\frac{L\bar{\sigma}^2}{n\varepsilon^2} + \frac{L\bar{\tau}}{p\varepsilon^{\frac{3}{2}}} + \frac{L}{p\varepsilon}\right)f_0\,, \quad (6)$$

*where $T$ is the number of iterations, $r_0 = \|\theta^{(0)} - \theta^\star\|_2^2$, $f_0 = f(\theta^{(0)}) - f^\star$ and $\mathcal{O}(\cdot)$ hides the numerical constants explicitly provided in the proof.*

We first note that our rates are consistent with those of the literature. Taking the complete graph allows getting rid of the middle terms and recover the rate of Parallel SGD [8], while with Assumption 5 we can bound $\bar{\tau}$ using Proposition 1 and recover the state-of-the-art bound of [17]. In addition to requiring less restrictive assumptions than those found in [17], the above rates exhibit sharper bounds. This comes notably from the fact that, in the heterogeneous setting $\bar{\tau}$ can be much smaller than $\sqrt{1-p}(\bar{\zeta} + \bar{\sigma})$ (see discussion in Section 3.1), and that now the topology influence the convergence rates via both the mixing parameter $p$ and $\bar{\tau}$.

## 4 Learning the Topology

In the previous rates (5) and (6), the smaller is $\bar{\tau}$, the fewer iterations are needed to reach an error $\varepsilon$. This motivates the idea of learning a *sparse* topology $W$ that *approximately* minimizes (an upper bound of) the neighborhood heterogeneity $H$ in order to control the trade-off between the convergence rate and the per-iteration communication complexity, see Equation (2). However, minimizing $H$ in the general setting appears to be challenging, as Equation (3) should hold for all $\theta \in \mathbb{R}^d$. Below, we focus on *classification with label skew*, and show that Equation (3) simplifies to a more tractable quantity.

### 4.1 Statistical Learning with Label Skew

Label skew is an important type of data heterogeneity in federated classification problems [16, 13, 2]. In this framework, each agent $i$ is associated with a random variable $Z_i = (X_i, Y_i) \sim \mathcal{D}_i$ where $X_i \in \mathbb{R}^q$ represents the feature vector and $Y_i \in [\![1, \ldots, K]\!]$ the associated class label. The agents aim to learn a classifier $h_\theta : \mathbb{R}^q \to [\![1, \ldots, K]\!]$ parameterized by $\theta \in \mathbb{R}^p$ such that $h_\theta(X_i)$ is a good predictor of $Y_i$ for all $i$. The heterogeneity of the distributions $\{\mathcal{D}_i\}_{i=1}^n$ comes only from a *difference in the label distribution* $P_i(Y)$ i.e. $\mathcal{D}_i = P_i(X, Y) = P(X|Y)P_i(Y)$. For simplicity, we assume that all agents use the same pointwise loss function ($F_i = F$ for all $i$), e.g., the cross-entropy. Under this framework, we can obtain an upper bound on $H$ that can effectively be minimized.

**Proposition 2** (Upper bound on $H$ under label skew). *Consider the statistical framework defined above and assume there exists $B > 0$ such that $\forall k = 1, \ldots, K$ and $\forall \theta \in \mathbb{R}^d$,*

$\|\mathbb{E}_X[\nabla F(\theta; X, Y)|Y = k] - \frac{1}{K}\sum_{k'=1}^{K}\mathbb{E}_X[\nabla F(\theta; X, Y)|Y = k']\|_2^2 \leq B$. *Then, denoting* $\pi_{jk} \triangleq P_j(Y = k)$, *we can bound the neighborhood heterogeneity* $H$ *in Equation* (3) *by:*

$$H \leq \frac{KB}{n}\sum_{k=1}^{K}\sum_{i=1}^{n}\Big(\sum_{j=1}^{n}W_{ij}\pi_{jk} - \frac{1}{n}\sum_{j=1}^{n}\pi_{jk}\Big)^2 + \frac{\sigma_{\max}^2}{n}\|W - \frac{1}{n}\mathbf{1}\mathbf{1}^\mathsf{T}\|_F^2 \, . \tag{7}$$

The proof is provided in Appendix D. Note that the assumption with $B$ corresponds to a bounded heterogeneity hypothesis at the class level (rather than at the agent level as in Assumption 5).

The upper bound (7) is quadratic in $W$ and is equal to 0 (minimum) when $W = \frac{\mathbf{1}\mathbf{1}^\mathsf{T}}{n}$, the complete topology with uniform weights. However, as already discussed, such a dense mixing matrix is impractical as it yields huge communication costs. We will show how the per-iteration communication complexity can be controlled while *approximately* minimizing the upper bound (7).

## 4.2 Optimization with the Frank-Wolfe Algorithm

In this section, we design an algorithm that finds a sparse approximate minimizer of the quantity (7). We focus on learning a single mixing matrix $W$ as a "pre-processing" step (before running D-SGD), and do so in a centralized manner. Specifically, we assume that a single party (which may be one of the agents, or a third-party) has access to the class proportions $\pi_{ik} = P_i(Y = k)$ for each agent $i$ and each class $k$. In practice, since each agent has access to its local dataset, it can compute these local proportions locally and share them without sharing the local data itself.

**Optimization problem.** Our objective is to learn a sparse mixing matrix $W$ which (approximately) minimizes the upper bound (7). Denoting by $\mathcal{S} \triangleq \{W \in [0, 1]^{n \times n} : W\mathbf{1} = \mathbf{1}, \ \mathbf{1}^\mathsf{T}W = \mathbf{1}^\mathsf{T}\}$ the set of doubly stochastic matrices, the optimization problem can be written as follows:

$$\min_{W \in \mathcal{S}} \Big\{g(W) \triangleq \frac{1}{n}\Big\|W\Pi - \frac{\mathbf{1}\mathbf{1}^\mathsf{T}}{n}\Pi\Big\|_F^2 + \frac{\lambda}{n}\Big\|W - \frac{\mathbf{1}\mathbf{1}^\mathsf{T}}{n}\Big\|_F^2\Big\}, \tag{8}$$

where $\Pi \in [0, 1]^{n \times K}$ contains the class proportions $\{\pi_{ik}\}$ and $\lambda > 0$ is a hyperparameter. To exactly match (7), $\lambda$ should be equal to $\frac{\sigma_{\max}^2}{KB}$ (but $\sigma_{\max}^2$ and $B$ are unknown in practice). Instead, we use $\lambda$ to control the bias-variance trade-off. As discussed in Section 3.1, the variance term is an upper bound of $1 - p$ with $p$ the mixing parameter of $W$. Therefore, $\lambda$ allows to tune a trade-off between the minimization of the bias due to label skew and the maximization of the mixing parameter of $W$.

**Algorithm.** We propose to find *sparse* approximations of (8) using a Frank-Wolfe (FW) algorithm, which is well-suited to learn a *sparse* parameter over convex hulls of finite set of atoms [15]. In our case, $\mathcal{S}$ corresponds to the convex hull of the set $\mathcal{A}$ of all permutation matrices [25, 32, 33].

FW applied to our specific framework is summarized in Algorithm 2 of Appendix A.2 and is referred to as Sparse Topology Learning with Frank-Wolfe (STL-FW). In a nutshell, at each iteration $t$ STL-FW finds the best (sparse) matrix $P \in \mathcal{A}$ in the opposite direction of the gradient $\nabla g(\widehat{W}^{(t)})$ and then performs a convex update of the current iterate $\widehat{W}^{(t)}$ with $P$. Crucially, STL-FW allows to control the sparsity of the final solution: at the end of the $l$-th iteration, each node will have at most $l$ in-neighbors and $l$ out-neighbors. The per-iteration communication complexity induced by the learned topology can thus be directly controlled by the number of iterations of our algorithm. The trade-off with the quality of the solution is quantified by the following theorem.

**Theorem 2.** *Consider the statistical setup presented in Section 4.1 and let* $\{\widehat{W}^{(l)}\}_{l=1}^{L}$ *be the sequence of mixing matrices generated by STL-SW (Appendix A.2). Then, at any iteration* $l = 1, \ldots, L$, *we have:*

$$g(\widehat{W}^{(l)}) \leq \frac{16}{l+2}\Big(\lambda + \frac{1}{n}\big\|\sum_{k=1}^{K}(\Pi_{:,k} - \overline{\Pi_{:,k}}\mathbf{1}) \cdot \Pi_{:,k}^\mathsf{T}\big\|_2^\star\Big), \tag{9}$$

*where* $\|\cdot\|_2^\star$ *stands for the nuclear norm, i.e., the sum of singular values. Furthermore, we have* $d_{max}^{in}(\widehat{W}^{(l)}) \leq l$ *and* $d_{max}^{out}(\widehat{W}^{(l)}) \leq l$, *resulting in a per-iteration complexity bounded by* $l$.

The above theorem shows that the objective $g$ decreases at a rate of $\mathcal{O}(1/l)$ as new connections between nodes are made. In general, we can bound (9) less tightly by $g(\widehat{W}^{(l)}) \leq \frac{16}{l+2}(\lambda + 1)$, which is *independent of the number of nodes* $n$. Recall that with $\lambda = \sigma_{\max}^2/KB$, the upper bound (7) over $H$ is exactly equal to $KB \cdot g(W)$. Therefore, the bound given in Theorem 2 also naturally provides a bound on the neighborhood heterogeneity $H$, which can be plugged in the rates of Theorem 1.

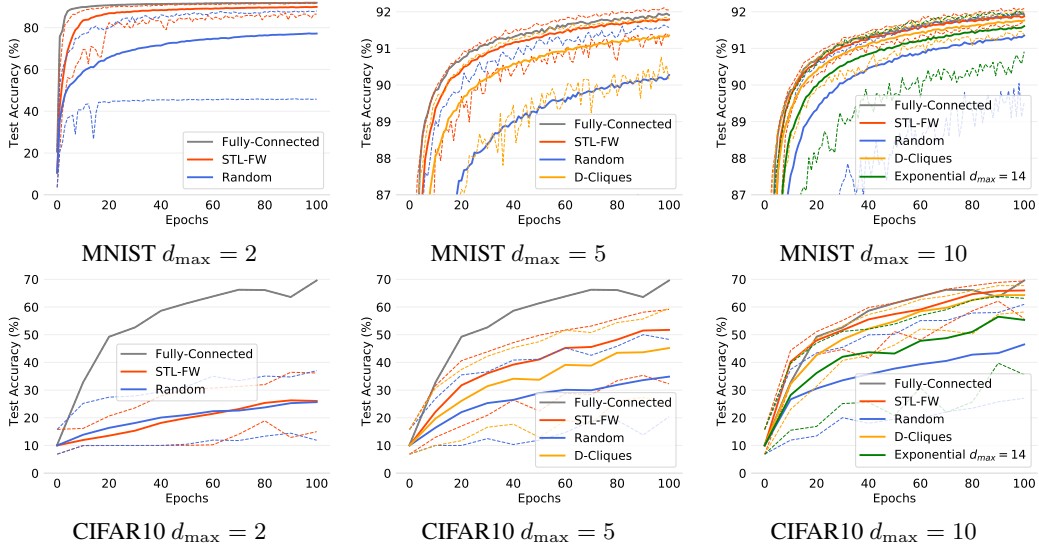

**Figure 1:** Convergence of D-SGD with STL-FW (our approach) and alternative topologies on real datasets under different communication budgets. The fully connected graph induces intractable communication costs but gives a performance upper bound, while the exponential graph is shown for $d_{\max} = 10$ but exceeds this budget.

## 5    Experiments

This section shows the practical usefulness of our topology learning method STL-SW. We call *communication budget* $d_{\max} = \max\{d_{\max}^{\text{in}}, d_{\max}^{\text{out}}\}$ the maximal number of neighbors a node can have in the used topologies, which controls the per-iteration communication complexity incurred by any node. Additional results on both synthetic and real-world data can be found in Appendix E.

**Setup.** We follow the setup in [2] and consider two classification tasks: a linear model on MNIST [9] and Group Normalized LeNet [13] on CIFAR10 [20]. In both cases, we partition the dataset on 100 nodes using the scheme introduced in [27], i.e. on average, nodes will have examples of two classes, but may have only 1 and up to 4. We re-use the hyper-parameters from [2]: learning rate of 0.1 and batch size of 128 for MNIST, and learning rate of 0.002 and batch size of 20 for CIFAR10. We compare D-SGD with STL-FW to other fixed topologies: (1) a *fully-connected graph* ($d_{\max} = 99$), which exhibits the fastest convergence speed, (2) a *random graph* with the same communication budget as STL-FW, (3) *D-Cliques* [2], also with the same budget, and (4) a deterministic *exponential graph* promoted in recent work [35] ($d_{\max} = 14$). Note that all competing topologies are data-independent, except D-Cliques. For all experiments with STL-FW, we use $\lambda = 0.1$ since, remarkably, its value does not significantly change the results (see Fig. 3 in Appendix E).

**Results.** Figure 1 shows our results for varying communication budget $d_{max}$: small (2), medium (5) and large (10). On MNIST, STL-FW converges faster than all competitors and quickly tends to the fully-connected topology as the budget $d_{\max}$ increases. Remarkably, STL-FW is already showing good performance at $d_{\max} = 2$, which is a very small budget that the other topologies (except the random one) cannot achieve. As expected, the two data-dependent topologies (D-Cliques and STL-FW) outperform the random topologies, including the exponential graph which has better connectivity ($d_{\max} = 14$) but does not compensate for the heterogeneity. The fact that STL-FW improves over D-Cliques may be explained by the fact that D-Cliques only compensate the heterogeneity (the bias term in $H$) without consideration for the overall connectivity (the variance term in $H$).

On CIFAR10, we see that $d_{\max} = 2$ is not sufficient to reach good performance. This can be explained by the increased complexity of the problem (nonconvex objective with a deep model), requiring larger communication budgets. This is in line with empirical results in prior work [19]. However, with slightly larger budgets i.e. $d_{\max} = 5$ and 10 ($d_{\max} = 3$ in Fig. 5, App. E), performance improves and the results are consistent with those on MNIST: STL-FW outperforms other sparse topologies and comes close to the performance of the fully connected topology for $d_{\max} = 10$. Overall, STL-FW provides better convergence speed than all tractable alternatives, with the additional ability to operate in low communication regimes (unlike D-Cliques and the exponential graph).

## 6    Conclusion

In this paper we showed, thanks to the new notion of neighborhood heterogeneity, that if chosen appropriately, the topology can compensate for the heterogeneity and speed up convergence. We also tackled the problem of learning such a good topology under data heterogeneity. To the best of our knowledge, our work is the first to provide a data-dependent approach, with explicit control on the trade-off between the communication cost and the convergence speed of D-SGD. We believe that our work paves the way for the design of other data-dependent topology learning techniques. One may for instance investigate different types of heterogeneity (beyond label skew), different knowledge assumptions (e.g., not knowing the proportions), and dynamic learning of the topology.

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

# Appendix

## A    Algorithms

### A.1    Decentralized Stochastic Gradient Descent

---

**Algorithm 1** Decentralized SGD [22, 17]

---

**Require:** Initialize $\forall i, \theta_i^{(0)} = \theta^{(0)} \in \mathbb{R}^d$, iterations $T$, stepsizes $\{\eta_t\}_{t=0}^{T-1}$, mixing $\{W^{(t)}\}_{t=0}^{T-1}$.
   **for** $t = 0, \ldots, T-1$ **do**
      **for** each node $i = 1, \ldots, n$ (in parallel) **do**
         Sample $Z_i^{(t)} \sim \mathcal{D}_i$
         $\theta_i^{(t+\frac{1}{2})} \leftarrow \theta_i^{(t)} - \eta_t \nabla F_i(\theta_i^{(t)}, Z_i^{(t)})$
         $\theta_i^{(t+1)} \leftarrow \sum_{j=1}^n W_{ij}^{(t)} \theta_j^{(t+\frac{1}{2})}$
      **end for**
   **end for**

---

### A.2    Sparse Topology Learning with Frank-Wolfe (STL-FW)

---

**Algorithm 2** Sparse Topology Learning with Frank-Wolfe (STL-FW)

---

**Require:** Initialization $\widehat{W}^{(0)} = I_n$, class proportions $\Pi \in [0, 1]^{n \times K}$ and hyperparameter $\lambda > 0$.
   **for** $l = 0, \ldots, L$ **do**
      $P^{(l+1)} = \arg\min_{P \in \mathcal{A}} \langle P, \nabla g(\widehat{W}^{(l)}) \rangle$              ▷ Find best permutation matrix
      $\gamma^{(l+1)} = \arg\min_{\gamma \in [0,1]} g\big((1-\gamma)\widehat{W}^{(l)} + \gamma P^{(l+1)}\big)$       ▷ Line-search
      $\widehat{W}^{(l+1)} = (1 - \gamma^{(l+1)})\widehat{W}^{(l)} + \gamma^{(l+1)} P^{(l+1)}$          ▷ Convex update
   **end for**

---

Starting from the identity matrix $\widehat{W}^{(0)} = I_n \in \mathcal{S}$, each iteration $l \geq 0$ consists of moving towards a feasible point $P^{(l+1)}$ that minimizes a linearization of $g$ at the current iterate $\widehat{W}^{(l)}$. As finding $P^{(l+1)}$ is a linear problem, solving it over $\mathcal{S}$ is equivalent to solving it over $\mathcal{A}$. Although $\mathcal{A}$ contains $n!$ elements, the linear program corresponds to the well-known *assignment problem* [5, 7] and can be solved efficiently with the Hungarian algorithm in time $\mathcal{O}(n^3)$ [25]. Note that the gradient needed to solve the assignment problem is given by

$$\nabla g(W) = \frac{2}{n} \sum_{k=1}^{K} (W\Pi_{:,k} - \overline{\Pi_{:,k}}\mathbf{1}) \cdot \Pi_{:,k}^{\mathsf{T}} + \frac{2}{n}\lambda \left(W - \frac{\mathbf{11}^{\mathsf{T}}}{n}\right) ,$$

where $\Pi_{:,k}$ corresponds to the $k$-th column of $\Pi$. The next iterate $\widehat{W}^{(l+1)}$ is then obtained as a convex combination of $P^{(l+1)}$ and $\widehat{W}^{(l)}$, and is thus guaranteed to be in $\mathcal{S}$. The optimal combining weight is computed by line-search, which has a closed-form solution since $g$ is quadratic.

Indeed, recall that we seek to solve:

$$\gamma^{\star} = \underset{\gamma \in [0,1]}{\arg\min} \ \left\{\tilde{g}(\gamma) \triangleq g\left((1-\gamma)W + \gamma P\right)\right\} ,$$

with

$$g(W) = \frac{1}{n}\left\|W\Pi - \frac{\mathbf{11}^{\mathsf{T}}}{n}\Pi\right\|_F^2 + \frac{\lambda}{n}\left\|W - \frac{\mathbf{11}^{\mathsf{T}}}{n}\right\|_F^2 .$$

The function $g$ being quadratic, the objective $\tilde{g}(\gamma)$ is also quadratic with respect to $\gamma$. Hence, it suffices to put the derivative $\tilde{g}'$ of $\tilde{g}$ equal to 0, and we get the closed-form solution:

$$\gamma^\star = \frac{\sum_{k=1}^{K}(\overline{\Pi_{:,k}}\mathbf{1} - W\Pi_{:,k})^\mathsf{T}(P - W)\cdot\Pi_{:,k} - \lambda\cdot\mathrm{tr}\left(\left(W - \frac{\mathbf{1}\mathbf{1}^\mathsf{T}}{n}\right)^\mathsf{T}(P-W)\right)}{\|(P-W)\Pi\|_F^2 + \lambda\|P - W\|_F^2}\,.$$

## B  Example of a sparse topology

Recall that we want to find an example where Assumption 5 is not verified while Assumption 4 is.

Let us consider $n$ nodes with $n$ an even number. For all $i = 1,\ldots,n$, assume $Z_i \sim \mathcal{N}(m,\tilde{\sigma}^2)$ if $i$ is odd and $Z_i \sim \mathcal{N}(-m,\tilde{\sigma}^2)$ if $i$ is even. Assume further that $\tilde{\sigma}^2 < +\infty$ but $m > 0$ can be asymptotically large. For all $i = 1,\ldots,n$ we fix $F_i(\theta, Z_i) = (\theta - Z_i)^2$, which corresponds to a simple mean estimation objective.

Consider a fixed mixing matrix $W$ associated with a ring topology that alternates between the two distributions. Specifically, for $i = 2,\ldots,n-1$ and $j = 1,\ldots,n$, we fix the weights as follows:

$$W_{ij} = \begin{cases} \frac{1}{2} & \text{if } j = i\,, \\ \frac{1}{4} & \text{if } j = i+1 \text{ or } j = i-1\,, \\ 0 & \text{otherwise}\,. \end{cases}$$

Moreover, we fix $W_{11} = W_{nn} = \frac{1}{2}$ and $W_{1n} = W_{n1} = \frac{1}{4}$.

With such parametrization we have $\nabla F_i(\theta, Z_i) = 2(\theta - Z_i)$ and therefore $\nabla f_i(\theta) = 2(\theta - m)$ if $i$ is odd and $\nabla f_i(\theta) = 2(\theta + m)$ if $i$ is even. Moreover, the gradient of the global objective is $\nabla f(\theta) = \frac{1}{n}\sum_i \nabla f_i(\theta) = 2\theta$ and the neighborhood averaging $\sum_j W_{ij}\nabla f_j(\theta) = 2\theta$ for all $i$.

We first verify that Assumptions 2 is satisfied:

$$\mathbb{E}\left[(\nabla F_i(\theta, Z_i) - \nabla f_i(\theta))^2\right] = \mathbb{E}\left[4(Z_i - \mathbb{E}Z_i)^2\right] = 4\tilde{\sigma}^2 < \infty\,.$$

Let us now find a bound $\bar{\tau}^2$ on the neighborhood heterogeneity $H$. Using a bias-variance decomposition, we have:

$$H = \frac{1}{n}\sum_{i=1}^{n}\mathbb{E}\left(\sum_{j=1}^{n}W_{ij}\nabla F_j(\theta) - \frac{1}{n}\sum_{j=1}^{n}\nabla F_j(\theta)\right)^2$$

$$= \frac{1}{n}\sum_{i=1}^{n}\left(\sum_{j=1}^{n}W_{ij}\nabla f_j(\theta) - \frac{1}{n}\sum_{j=1}^{n}\nabla f_j(\theta)\right)^2 + \frac{1}{n}\sum_{i=1}^{n}\mathbb{E}\left(\sum_{j=1}^{n}(W_{ij} - \frac{1}{n})(\nabla f_j(\theta) - \nabla F_j(\theta))\right)^2$$

$$= \frac{1}{n}\sum_{i=1}^{n}(2\theta - 2\theta)^2 + \frac{1}{n}\sum_{i=1}^{n}\sum_{j=1}^{n}(W_{ij} - \frac{1}{n})^2\mathbb{E}(\nabla f_j(\theta) - \nabla F_j(\theta))^2$$

$$= 0 + 4\tilde{\sigma}^2\frac{1}{n}\sum_{i=1}^{n}\sum_{j=1}^{n}(W_{ij} - \frac{1}{n})^2 \le 4\tilde{\sigma}^2\,.$$

The third equality was obtained thanks to the fact that $\mathbb{E}[\nabla f_j(\theta) - \nabla F_j(\theta)] = 0$. This result shows that Assumption 4 is verified with $\bar{\tau}^2 = 4\tilde{\sigma}^2 < \infty$.

On the contrary, since $m$ can be arbitrary large, Assumption 5 is not verified. Indeed:

$$\frac{1}{n}\sum_{i=1}^{n}\left(\nabla f_i(\theta) - \frac{1}{n}\sum_{j=1}^{n}\nabla f_j(\theta)\right)^2 = \frac{1}{n}\sum_{i=1}^{n}(2m)^2$$

$$= \frac{4m^2}{n} \triangleq \bar{\zeta}^2 \xrightarrow[m\to\infty]{} +\infty\,.$$

**Remark.** At first sight, one may wonder why the local variance term $\tilde{\sigma}^2$ appears in $\bar{\tau}^2$ but not in $\bar{\zeta}^2$. This is because we chose to define neighborhood heterogeneity in expectation with respect to the

pointwise loss functions $F_1, \ldots, F_n$, resulting in a bias-variance decomposition (see Eq. 4) which is the relevant quantity to optimize when learning the topology in Section 4. In contrast, following the convention used in previous work, local heterogeneity is defined with respect to the local objectives $f_1, \ldots, f_n$ and thus only measures a bias term, while the variance term is accounted separately by Assumption 2. Since the variance terms are the same in both settings, the difference is in how the bias term is measured (at the node level or at the neighborhood level): in the example above, it is equal to $\frac{4m^2}{n}$ for local heterogeneity while it is equal to 0 for neighborhood heterogeneity (see the above calculation of $H$).

## C  Proof of Theorem 1

### C.1  Notations and Overview

We start by re-writing the updates of D-SGD (Algorithm 1) in matrix form.

Let $\Theta^{(t)} \triangleq \left(\theta_1^{(t)}, \ldots, \theta_n^{(t)}\right) \in \mathbb{R}^{d \times n}$ be the matrix that contains the parameter vectors of all nodes at time $t$. Denote by $\nabla F(\Theta^{(t)}, Z^{(t)}) \triangleq \left(\nabla F_1(\theta_1^{(t)}, Z_1^{(t)}), \ldots, \nabla F_n(\theta_n^{(t)}, Z_n^{(t)})\right) \in \mathbb{R}^{d \times n}$ the matrix containing all stochastic gradients at time $t$. The D-SGD update at time $t$ can then be written as:

$$\Theta^{(t+1)} = \left(\Theta^{(t)} - \eta_t \nabla F(\Theta^{(t)}, Z^{(t)})\right) W^{(t)\mathsf{T}} .$$

In the following, we denote $\overline{\Theta}^{(t)} \triangleq \left(\bar{\theta}^{(t)}, \ldots, \bar{\theta}^{(t)}\right) = \Theta^{(t)} \cdot \frac{1}{n}\mathbf{1}\mathbf{1}^\mathsf{T}$.

The proof follows the classical steps found in the literature (see e.g. [17, 29]). The main difference resides in how the consensus term $\|\Theta^{(t)} - \overline{\Theta}^{(t)}\|_F^2$ is controlled across iterations (Lemma 3). The proof is organized as follows.

**Convex case.**

1. Lemma 1 provides a descent recursion that allows to control the decreasing of the term $\left\|\bar{\theta}^{(t)} - \theta^\star\right\|^2$. The proof closely follows the one of [17, 29].

2. In Lemma 3, the consensus term $\|\Theta^{(t)} - \overline{\Theta}^{(t)}\|_F^2$, which appears in the result of Lemma 1, is upper-bounded. The resulting upper-bound exhibits our new quantity $\bar{\tau}^2$ (an upper bound on neighborhood heterogeneity).

3. Corollary 1 uses the previous lemma to bound $\frac{1}{T+1}\sum_{t=0}^{T} \|\Theta^{(t)} - \overline{\Theta}^{(t)}\|_F^2$.

4. Lemma 4 provides an upper-bound on the error term with the following form:

$$\frac{1}{T+1}\sum_{t=0}^{T} \mathbb{E}(f(\bar{\theta}^{(t)}) - f^\star) \leq 2\left(\frac{br_0}{T+1}\right)^{\frac{1}{2}} + 2e^{\frac{1}{3}}\left(\frac{r_0}{T+1}\right)^{\frac{2}{3}} + \frac{dr_0}{T+1},$$

where $b = \frac{\bar{\sigma}^2}{n}$, $e = \frac{36L\bar{\tau}^2}{p^2}$, $d = \frac{8L}{p}$ and $r_0 = \|\theta^{(0)} - \theta^\star\|_2^2$.

5. To get the final rate of Theorem 1, it suffices to find $T$ such that each term in the right-hand side of the previous equation in bounded by $\frac{\varepsilon}{3}$.

   - $2\left(\frac{br_0}{T+1}\right)^{\frac{1}{2}} \leq \frac{\varepsilon}{3} \iff \frac{36br_0}{\varepsilon^2} \leq T+1 \iff \frac{36\bar{\sigma}^2 r_0}{n\varepsilon^2} \leq T+1,$

   - $2e^{\frac{1}{3}}\left(\frac{r_0}{T+1}\right)^{\frac{2}{3}} \leq \frac{\varepsilon}{3} \iff \frac{e^{\frac{1}{2}}6^{\frac{3}{2}}r_0}{\varepsilon^{\frac{3}{2}}} \leq T+1 \iff \frac{6^{\frac{5}{2}}\sqrt{L}\bar{\tau}r_0}{p\varepsilon^{\frac{3}{2}}} \leq T+1,$

   - $\frac{dr_0}{T+1} \leq \frac{\varepsilon}{3} \iff \frac{3dr_0}{\varepsilon} \leq T+1 \iff \frac{24Lr_0}{p\varepsilon} \leq T+1.$

   In particular, it suffices to take

$$T \geq \frac{36\bar{\sigma}^2 r_0}{n\varepsilon^2} + \frac{89\sqrt{L}\bar{\tau}r_0}{p\varepsilon^{\frac{3}{2}}} + \frac{24Lr_0}{p\varepsilon} = \mathcal{O}\left(\frac{\bar{\sigma}^2}{n\varepsilon^2} + \frac{\sqrt{L}\bar{\tau}}{p\varepsilon^{\frac{3}{2}}} + \frac{L}{p\varepsilon}\right)r_0 ,$$

   in order to have all three terms bounded by $\frac{\varepsilon}{3}$, and obtain the final result.

**Non-convex case.** The proof is similar to the convex one: it only differs in the descent lemmas that are used.

1. Lemma 2 provides the descent lemma for the non-convex scenario.
2. The consensus term is bounded using the same results as in the convex case, i.e., with Lemma 3 and Corollary 1.
3. Lemma 5 provides an upper-bound on the error term with the following form:

$$\frac{1}{T+1}\sum_{t=0}^{T}\mathbb{E}\left\|\nabla f(\bar{\theta}^{(t)})\right\|_2^2 \leq 2\left(\frac{4bf_0}{T+1}\right)^{\frac{1}{2}} + 2e^{\frac{1}{3}}\left(\frac{4f_0}{T+1}\right)^{\frac{2}{3}} + \frac{4df_0}{T+1},$$

where $b = \frac{2L\bar{\sigma}^2}{n}$, $e = \frac{96L^2\bar{\tau}^2}{p^2}$, $d = \frac{8L}{p}$ and $f_0 = f(\theta^{(0)}) - f^\star$.

4. We bound in each term of the previous equation by $\frac{\varepsilon}{3}$ and get the sufficient condition:

$$T \geq \frac{288L\bar{\sigma}^2 f_0}{n\varepsilon^2} + \frac{576L\bar{\tau} f_0}{p\varepsilon^{\frac{3}{2}}} + \frac{96Lf_0}{p\varepsilon} = \mathcal{O}\left(\frac{L\bar{\sigma}^2}{n\varepsilon^2} + \frac{L\bar{\tau}}{p\varepsilon^{\frac{3}{2}}} + \frac{L}{p\varepsilon}\right)f_0\,.$$

## C.2 Preliminaries and Useful Results

**Property 1** (Averaging preservation). *Let $W \in \mathbb{R}^{n\times n}$ be a mixing matrix and $\Theta$ be any matrix in $\mathbb{R}^{d\times n}$. Then, $W$ preserves averaging:*

$$(\Theta W)\frac{\mathbf{1}\mathbf{1}^\mathsf{T}}{n} = \Theta\frac{\mathbf{1}\mathbf{1}^\mathsf{T}}{n} = \overline{\Theta} \tag{10}$$

**Property 2** (Implications of $L$-smoothness and convexity).

- *If we assume convexity, we have for all $i \in [\![1,\dots,n]\!]$:*
$$\langle\nabla f_i(\tilde{\theta}), \tilde{\theta} - \theta\rangle \geq f_i(\tilde{\theta}) - f_i(\theta). \tag{11}$$

- *Under Assumption 1 ($L$-smoothness), it holds for all $i \in [\![1,\dots,n]\!]$:*
$$F_i(\theta, Z) \leq F_i(\tilde{\theta}, Z) + \langle\nabla F_i(\tilde{\theta}, Z), \theta - \tilde{\theta}\rangle + \frac{L}{2}\|\theta - \tilde{\theta}\|_2^2, \qquad \forall\theta, \tilde{\theta} \in \mathbb{R}^d, Z \in \theta_i. \tag{12}$$

  *Taking the expectation of the previous equation, we also have:*
$$f_i(\theta) \leq f_i(\tilde{\theta}) + \langle\nabla F(\tilde{\theta}), \theta - \tilde{\theta}\rangle + \frac{L}{2}\|\theta - \tilde{\theta}\|_2^2, \qquad \forall\theta, \tilde{\theta} \in \mathbb{R}^d. \tag{13}$$

- *If we further assume that the $F_i$'s are convex, Assumption 1 also implies $\forall\theta, \tilde{\theta} \in \mathbb{R}^d, Z \in \theta_i$:*
$$\|\nabla f_i(\theta) - \nabla f_i(\tilde{\theta})\|_2 \leq L\|\theta - \tilde{\theta}\|_2, \tag{14}$$
$$\|\nabla f_i(\theta) - \nabla f_i(\tilde{\theta})\|_2^2 \leq 2L\left(f_i(\theta) - f_i(\tilde{\theta}) - \langle\nabla f_i(\tilde{\theta}), \theta - \tilde{\theta}\rangle\right), \tag{15}$$
$$\|\nabla F_i(\theta, Z) - \nabla F_i(\tilde{\theta}, Z)\|_2^2 \leq 2L\left(F_i(\theta, Z) - F_i(\tilde{\theta}, Z) - \langle\nabla F_i(\tilde{\theta}, Z), \theta - \tilde{\theta}\rangle\right). \tag{16}$$

These results can be found in many convex optimization books and papers, e.g. in [4].

**Property 3** (Norm inequalities).

- *For a set of vectors $\{a_i\}_{i=1}^n$ such that $a_i \in \mathbb{R}^d$,*

$$\left\|\sum_{i=1}^n a_i\right\|_2^2 \leq n\sum_{i=1}^n\|a_i\|_2^2. \tag{17}$$

- *For two vectors $a, b \in \mathbb{R}^d$,*
$$\|a + b\|_2^2 \leq (1+\alpha)\|a\|_2^2 + (1+\alpha^{-1})\|b\|_2^2, \qquad \forall\alpha > 0. \tag{18}$$

- *For two vectors $a, b \in \mathbb{R}^d$,*
$$2\langle a, b\rangle \leq \alpha\|a\|_2^2 + \alpha^{-1}\|b\|_2^2, \qquad \forall\alpha > 0. \tag{19}$$

### C.3 Needed Lemmas

In the following we denote by $\mathcal{F}_t = \sigma(Z^{(k)} | k \leq t)$ the natural filtration with respect to $Z^{(t)} = (Z_1^{(t)}, \ldots, Z_n^{(t)})$. Remark that $\forall i = 1, \ldots, n$ the iterates $\theta_i^{(t+1)}$ and $\bar{\theta}^{(t+1)}$ are in particular $\mathcal{F}_t$-measurable.

**Lemma 1** (Descent Lemma - Convex case)**.** *Consider the setting of Theorem 1 and let $\eta_t \leq \frac{1}{4L}$, then we almost surely have:*

$$\mathbb{E}_{Z^{(t)} | \mathcal{F}_{t-1}} \left\| \bar{\theta}^{(t+1)} - \theta^\star \right\|^2 \leq \left\| \bar{\theta}^{(t)} - \theta^\star \right\|^2 + \frac{\eta_t^2 \bar{\sigma}^2}{n} - \eta_t \left( f(\bar{\theta}^{(t)}) - f^\star \right) + \frac{3L}{2n} \eta_t \left\| \Theta^{(t)} - \overline{\Theta}^{(t)} \right\|_F^2, \tag{20}$$

*where $\mathbb{E}_{Z^{(t)} | \mathcal{F}_{t-1}}$ stands for the conditional expectation $\mathbb{E}_{Z^{(t)}}[\cdot | \mathcal{F}_{t-1}]$.*

*Proof.* The proof closely follows the one in [17]. Using the recursion of D-SGD and since all mixing matrices are doubly stochastic and preserve the average (Proposition 1) we have:

$$\|\bar{\theta}^{(t+1)} - \theta^\star\|^2 = \left\| \bar{\theta}^{(t)} - \frac{\eta_t}{n} \sum_{i=1}^n \nabla F_i(\theta_i^{(t)}, Z_i^{(t)}) - \theta^\star \right\|^2$$

$$= \left\| \bar{\theta}^{(t)} - \theta^\star - \frac{\eta_t}{n} \sum_{i=1}^n \nabla f_i(\theta_i^{(t)}) + \frac{\eta_t}{n} \sum_{i=1}^n \nabla f_i(\theta_i^{(t)}) - \frac{\eta_t}{n} \sum_{i=1}^n \nabla F_i(\theta_i^{(t)}, Z_i^{(t)}) \right\|^2$$

$$= \left\| \bar{\theta}^{(t)} - \theta^\star - \frac{\eta_t}{n} \sum_{i=1}^n \nabla f_i(\theta_i^{(t)}) \right\|^2 + \eta_t^2 \left\| \frac{1}{n} \sum_{i=1}^n \nabla f_i(\theta_i^{(t)}) - \frac{1}{n} \sum_{i=1}^n \nabla F_i(\theta_i^{(t)}, Z_i^{(t)}) \right\|^2$$

$$+ 2 \left\langle \bar{\theta}^{(t)} - \theta^\star - \frac{\eta_t}{n} \sum_{i=1}^n \nabla f_i(\theta_i^{(t)}), \frac{\eta_t}{n} \sum_{i=1}^n \nabla f_i(\theta_i^{(t)}) - \frac{\eta_t}{n} \sum_{i=1}^n \nabla F_i(\theta_i^{(t)}, Z_i^{(t)}) \right\rangle.$$

Passing to the conditional expectation, the last term (the inner product) is equal to $0$. This comes from the fact that $\mathbb{E}_{Z_i^{(t)} | \mathcal{F}_{t-1}} [\nabla F_i(\theta_i^{(t)}, Z_i^{(t)})] = \nabla f_i(\theta_i^{(t)})$. We therefore need to bound the first two terms in the conditional expectation.

The second one can easily be bounded using Assumption 2:

$$\eta_t^2 \mathbb{E}_{Z^{(t)} | \mathcal{F}_{t-1}} \left\| \frac{1}{n} \sum_{i=1}^n \nabla f_i(\theta_i^{(t)}) - \frac{1}{n} \sum_{i=1}^n \nabla F_i(\theta_i^{(t)}, Z_i^{(t)}) \right\|^2$$

$$= \frac{\eta_t^2}{n^2} \mathbb{E}_{Z^{(t)} | \mathcal{F}_{t-1}} \left\| \sum_{i=1}^n (\nabla f_i(\theta_i^{(t)}) - \nabla F_i(\theta_i^{(t)}, Z_i^{(t)})) \right\|^2$$

$$= \frac{\eta_t^2}{n^2} \sum_{i=1}^n \mathbb{E}_{Z_i^{(t)} | \mathcal{F}_{t-1}} \left\| \nabla f_i(\theta_i^{(t)}) - \nabla F_i(\theta_i^{(t)}, Z_i^{(t)}) \right\|^2$$

$$\overset{(A.2)}{\leq} \frac{\eta_t^2 \bar{\sigma}^2}{n},$$

where the second equality was obtained using the identity $\mathbb{E} \|\sum_i Y_i\|_2^2 = \sum_i \mathbb{E} \|Y_i\|_2^2$ when $Y_i$ are independent and $\mathbb{E} Y_i = 0$.

Now that the second term is bounded, we can move to the first one. Because $\theta_i^{(t)}$ and $\bar{\theta}^{(t)}$ are $\mathcal{F}_{t-1}$-measurable, we have

$$\mathbb{E}_{Z^{(t)} | \mathcal{F}_{t-1}} \left\| \bar{\theta}^{(t)} - \theta^\star - \frac{\eta_t}{n} \sum_{i=1}^n \nabla f_i(\theta_i^{(t)}) \right\|^2 = \left\| \bar{\theta}^{(t)} - \theta^\star - \frac{\eta_t}{n} \sum_{i=1}^n \nabla f_i(\theta_i^{(t)}) \right\|^2$$

$$= \left\| \bar{\theta}^{(t)} - \theta^\star \right\|^2 + \eta_t^2 \underbrace{\left\| \frac{1}{n} \sum_{i=1}^n \nabla f_i(\theta_i^{(t)}) \right\|^2}_{T_1} - 2\eta_t \underbrace{\left\langle \bar{\theta}^{(t)} - \theta^\star, \frac{1}{n} \sum_{i=1}^n \nabla f_i(\theta_i^{(t)}) \right\rangle}_{T_2}.$$

In order to bound $T_1$, recall that by definition $\frac{1}{n} \sum_i \nabla f_i(\theta^\star) = 0$, therefore:

$$
\begin{aligned}
T_1 &= \left\| \frac{1}{n} \sum_{i=1}^n (\nabla f_i(\theta_i^{(t)}) - \nabla f_i(\bar{\theta}^{(t)}) + \nabla f_i(\bar{\theta}^{(t)}) - \nabla f_i(\theta^\star)) \right\|^2 \\
&\overset{(17)}{\leq} 2 \left\| \frac{1}{n} \sum_{i=1}^n (\nabla f_i(\theta_i^{(t)}) - \nabla f_i(\bar{\theta}^{(t)})) \right\|^2 + 2 \left\| \frac{1}{n} \sum_{i=1}^n (\nabla f_i(\bar{\theta}^{(t)}) - \nabla f_i(\theta^\star)) \right\|^2 \\
&\overset{(17)}{\leq} \frac{2}{n} \sum_{i=1}^n \left\| \nabla f_i(\theta_i^{(t)}) - \nabla f_i(\bar{\theta}^{(t)}) \right\|^2 + \frac{2}{n} \sum_{i=1}^n \left\| \nabla f_i(\bar{\theta}^{(t)}) - \nabla f_i(\theta^\star) \right\|^2 \\
&\overset{(14)(15)}{\leq} \frac{2L^2}{n} \sum_{i=1}^n \left\| \theta_i^{(t)} - \bar{\theta}^{(t)} \right\|^2 + \frac{4L}{n} \sum_{i=1}^n \left( f_i(\bar{\theta}^{(t)}) - f_i(\theta^\star) - \left\langle \nabla f_i(\theta^\star), \bar{\theta}^{(t)} - \theta^\star \right\rangle \right) \\
&= \frac{2L^2}{n} \sum_{i=1}^n \left\| \theta_i^{(t)} - \bar{\theta}^{(t)} \right\|^2 + \frac{4L}{n} \sum_{i=1}^n \left( f_i(\bar{\theta}^{(t)}) - f_i(\theta^\star) \right) - 4L \underbrace{\left\langle \frac{1}{n} \sum_{i=1}^n \nabla f_i(\theta^\star), \bar{\theta}^{(t)} - \theta^\star \right\rangle}_{=0} \\
&= \frac{2L^2}{n} \sum_{i=1}^n \left\| \theta_i^{(t)} - \bar{\theta}^{(t)} \right\|^2 + 4L \left( f(\bar{\theta}^{(t)}) - f^\star \right).
\end{aligned}
$$

Finally, we have to bound $T_2$:

$$
\begin{aligned}
-T_2 &= -\frac{2\eta_t}{n} \sum_{i=1}^n \left\langle \bar{\theta}^{(t)} - \theta^\star, \nabla f_i(\theta_i^{(t)}) \right\rangle \\
&= -\frac{2\eta_t}{n} \sum_{i=1}^n \left[ \left\langle \bar{\theta}^{(t)} - \theta_i^{(t)}, \nabla f_i(\theta_i^{(t)}) \right\rangle + \left\langle \theta_i^{(t)} - \theta^\star, \nabla f_i(\theta_i^{(t)}) \right\rangle \right] \\
&\overset{(13)(11)}{\leq} -\frac{2\eta_t}{n} \sum_{i=1}^n \left[ f_i(\bar{\theta}^{(t)}) - f_i(\theta_i^{(t)}) - \frac{L}{2} \|\bar{\theta}^{(t)} - \theta_i^{(t)}\|_2^2 + f_i(\theta_i^{(t)}) - f_i(\theta^\star) \right] \\
&= -2\eta_t \left( f(\bar{\theta}^{(t)}) - f(\theta^\star) \right) + \frac{L\eta_t}{n} \sum_{i=1}^n \|\bar{\theta}^{(t)} - \theta_i^{(t)}\|_2^2 \\
&= -2\eta_t \left( f(\bar{\theta}^{(t)}) - f^\star \right) + \frac{L\eta_t}{n} \|\overline{\Theta}^{(t)} - \Theta^{(t)}\|_F^2.
\end{aligned}
$$

Combining all previous results, we get:

$$
\begin{aligned}
\mathbb{E}_{Z^{(t)}|\mathcal{F}_{t-1}} \left\| \bar{\theta}^{(t+1)} - \theta^\star \right\|^2 \leq{} & \left\| \bar{\theta}^{(t)} - \theta^\star \right\|^2 + \frac{\eta_t^2 \bar{\sigma}^2}{n} + \frac{L\eta_t}{n} (2L\eta_t + 1) \|\overline{\Theta}^{(t)} - \Theta^{(t)}\|_F^2 \\
& + 2\eta_t (2L\eta_t - 1) \left( f(\bar{\theta}^{(t)}) - f^\star \right).
\end{aligned}
$$

Since, by hypothesis, $\eta_t \leq \frac{1}{4L}$, we have $2L\eta_t + 1 \leq \frac{3}{2}$ and $2L\eta_t - 1 \leq -\frac{1}{2}$, which concludes the proof. $\qquad\square$

**Lemma 2** (Descent Lemma - Non-convex case). *Consider the setting of Theorem 1 and let $\eta_t \leq \frac{1}{4L}$, then we almost surely have:*

$$
\mathbb{E}_{Z^{(t)}|\mathcal{F}_{t-1}} f(\bar{\theta}^{(t+1)}) - f^\star \leq f(\bar{\theta}^{(t)}) - f^\star - \frac{\eta_t}{4} \|\nabla f(\bar{\theta}^{(t)})\|_2^2 + \frac{L^2}{n} \eta_t \left\| \Theta^{(t)} - \overline{\Theta}^{(t)} \right\|_F^2 + \frac{L\bar{\sigma}^2}{2n} \eta_t^2. \quad (21)
$$

*Proof.* The proof adapts the one of Lemma 10 in [17] to our setting.

$$\mathbb{E}_{Z^{(t)}|\mathcal{F}_{t-1}} f(\bar{\theta}^{(t+1)}) = \mathbb{E}_{Z^{(t)}|\mathcal{F}_{t-1}} f\left(\bar{\theta}^{(t)} - \frac{\eta_t}{n} \sum_{i=1}^{n} \nabla F_i(\theta_i^{(t)}, Z_i^{(t)})\right)$$

$$\overset{(13)}{\leq} f(\bar{\theta}^{(t)}) - \mathbb{E}_{Z^{(t)}|\mathcal{F}_{t-1}} \left\langle \nabla f(\bar{\theta}^{(t)}), \frac{\eta_t}{n} \sum_{i=1}^{n} \nabla F_i(\theta_i^{(t)}, Z_i^{(t)}) \right\rangle$$

$$+ \frac{L}{2} \mathbb{E}_{Z^{(t)}|\mathcal{F}_{t-1}} \left\| \frac{\eta_t}{n} \sum_{i=1}^{n} \nabla F_i(\theta_i^{(t)}, Z_i^{(t)}) \right\|_2^2$$

$$= f(\bar{\theta}^{(t)}) \underbrace{- \left\langle \nabla f(\bar{\theta}^{(t)}), \frac{\eta_t}{n} \sum_{i=1}^{n} \nabla f_i(\theta_i^{(t)}) \right\rangle}_{\triangleq T_4} + \underbrace{\frac{L\eta_t^2}{2} \mathbb{E}_{Z^{(t)}|\mathcal{F}_{t-1}} \left\| \frac{1}{n} \sum_{i=1}^{n} \nabla F_i(\theta_i^{(t)}, Z_i^{(t)}) \right\|_2^2}_{\triangleq T_5}$$

Adding and subtracting $\eta_t \nabla f(\bar{\theta}^{(t)})$ in $T_4$, we have

$$T_4 = -\eta_t \left\| \nabla f(\bar{\theta}^{(t)}) \right\|_2^2 + \frac{\eta_t}{n} \sum_{i=1}^{n} \left\langle \nabla f(\bar{\theta}^{(t)}), \nabla f_i(\bar{\theta}^{(t)}) - \nabla f_i(\theta_i^{(t)}) \right\rangle$$

$$\overset{(19),\alpha=1}{\leq} -\eta_t \left\| \nabla f(\bar{\theta}^{(t)}) \right\|_2^2 + \frac{\eta_t}{2} \left\| \nabla f(\bar{\theta}^{(t)}) \right\|_2^2 + \frac{\eta_t}{2n} \sum_{i=1}^{n} \left\| \nabla f_i(\bar{\theta}^{(t)}) - \nabla f_i(\theta_i^{(t)}) \right\|_2^2$$

$$\overset{(14)}{\leq} -\frac{\eta_t}{2} \left\| \nabla f(\bar{\theta}^{(t)}) \right\|_2^2 + \frac{L^2\eta_t}{2n} \sum_{i=1}^{n} \left\| \bar{\theta}^{(t)} - \theta_i^{(t)} \right\|_2^2 .$$

Let us now bound the term $T_5$:

$$T_5 = \mathbb{E}_{Z^{(t)}|\mathcal{F}_{t-1}} \left\| \frac{1}{n} \sum_{i=1}^{n} \nabla F_i(\theta_i^{(t)}, Z_i^{(t)}) - \frac{1}{n} \sum_{i=1}^{n} \nabla f_i(\theta_i^{(t)}) + \frac{1}{n} \sum_{i=1}^{n} \nabla f_i(\theta_i^{(t)}) \right\|_2^2$$

$$= \frac{1}{n^2} \mathbb{E}_{Z^{(t)}|\mathcal{F}_{t-1}} \left\| \sum_{i=1}^{n} \nabla F_i(\theta_i^{(t)}, Z_i^{(t)}) - \sum_{i=1}^{n} \nabla f_i(\theta_i^{(t)}) \right\|_2^2 + \left\| \frac{1}{n} \sum_{i=1}^{n} \nabla f_i(\theta_i^{(t)}) \right\|_2^2$$

$$\overset{(A.2)}{=} \frac{\bar{\sigma}^2}{n} + \left\| \frac{1}{n} \sum_{i=1}^{n} \nabla f_i(\theta_i^{(t)}) - \nabla f(\bar{\theta}^{(t)}) + \nabla f(\bar{\theta}^{(t)}) \right\|_2^2$$

$$\overset{(17)}{\leq} \frac{\bar{\sigma}^2}{n} + 2 \left\| \frac{1}{n} \sum_{i=1}^{n} \nabla f_i(\theta_i^{(t)}) - \nabla f(\bar{\theta}^{(t)}) \right\|_2^2 + 2 \left\| \nabla f(\bar{\theta}^{(t)}) \right\|_2^2$$

$$\overset{(17)}{\leq} \frac{\bar{\sigma}^2}{n} + \frac{2}{n} \sum_{i=1}^{n} \left\| \nabla f_i(\theta_i^{(t)}) - \nabla f_i(\bar{\theta}^{(t)}) \right\|_2^2 + 2 \left\| \nabla f(\bar{\theta}^{(t)}) \right\|_2^2$$

$$\overset{(14)}{\leq} \frac{\bar{\sigma}^2}{n} + \frac{2L^2}{n} \sum_{i=1}^{n} \left\| \theta_i^{(t)} - \bar{\theta}^{(t)} \right\|_2^2 + 2 \left\| \nabla f(\bar{\theta}^{(t)}) \right\|_2^2 .$$

Next, plugging $T_4$ and $T_5$ into the first inequality, we have:

$$\mathbb{E}_{Z^{(t)}|\mathcal{F}_{t-1}} f(\bar{\theta}^{(t+1)})$$

$$\leq f(\bar{\theta}^{(t)}) - \eta_t \left( \frac{1}{2} - L\eta_t \right) \left\| \nabla f(\bar{\theta}^{(t)}) \right\|_2^2 + \left( \frac{L^2\eta_t}{2n} + \frac{L^3\eta_t^2}{n} \right) \left\| \Theta^{(t)} - \overline{\Theta}^{(t)} \right\|_F^2 + \frac{L\bar{\sigma}^2}{2n} \eta_t^2 .$$

Since by hypothesis $\eta_t \leq \frac{1}{4L}$, we have $\frac{1}{2} - L\eta_t \geq \frac{1}{4}$ and $\frac{L^2\eta_t}{2n} + \frac{L^3\eta_t^2}{n} \leq \frac{L^2\eta_t}{n}$, we therefore get

$$\mathbb{E}_{Z^{(t)}|\mathcal{F}_{t-1}} f(\bar{\theta}^{(t+1)}) \leq f(\bar{\theta}^{(t)}) - \frac{\eta_t}{4} \left\| \nabla f(\bar{\theta}^{(t)}) \right\|_2^2 + \frac{L^2 \eta_t}{n} \left\| \Theta^{(t)} - \overline{\Theta}^{(t)} \right\|_F^2 + \frac{L\bar{\sigma}^2}{2n} \eta_t^2 .$$

Subtracting each side of the equation by $f^\star$, we obtain the final result. $\qquad\square$

**Lemma 3** (Consensus Control). *Consider the setting of Theorem 1 and let $\eta_t \leq \frac{p}{8L}$, then:*

$$\mathbb{E} \left\| \Theta^{(t)} - \overline{\Theta}^{(t)} \right\|_F^2 \leq (1 - \frac{p}{4}) \mathbb{E} \left\| \Theta^{(t-1)} - \overline{\Theta}^{(t-1)} \right\|_F^2 + \frac{6n\bar{\tau}^2}{p} \eta_{t-1}^2. \qquad (22)$$

*Proof.* For any $\alpha > 0$, we have:

$$
\begin{aligned}
\mathbb{E} \left\| \Theta^{(t)} - \overline{\Theta}^{(t)} \right\|_F^2 &= \mathbb{E} \left\| \Theta^{(t)} \left( I - \frac{\mathbf{1}\mathbf{1}^\mathsf{T}}{n} \right) \right\|_F^2 \\
&= \mathbb{E} \left\| \left( \Theta^{(t-1)} - \eta_{t-1} \nabla F(\Theta^{(t-1)}, Z^{(t-1)}) \right) W^{(t-1)\mathsf{T}} \left( I - \frac{\mathbf{1}\mathbf{1}^\mathsf{T}}{n} \right) \right\|_F^2 \\
&\overset{(10)}{=} \mathbb{E} \left\| \left( \Theta^{(t-1)} - \eta_{t-1} \nabla F(\Theta^{(t-1)}, Z^{(t-1)}) \right) \left( W^{(t-1)\mathsf{T}} - \frac{\mathbf{1}\mathbf{1}^\mathsf{T}}{n} \right) \right\|_F^2 \\
&\overset{(18)}{\leq} (1 + \alpha) \mathbb{E} \left\| \Theta^{(t-1)} \left( W^{(t-1)\mathsf{T}} - \frac{\mathbf{1}\mathbf{1}^\mathsf{T}}{n} \right) \right\|_F^2 \\
&\quad + (1 + \alpha^{-1}) \eta_{t-1}^2 \underbrace{\mathbb{E} \left\| \nabla F(\Theta^{(t-1)}, Z^{(t-1)}) \left( W^{(t-1)\mathsf{T}} - \frac{\mathbf{1}\mathbf{1}^\mathsf{T}}{n} \right) \right\|_F^2}_{T_3} \\
&\overset{(A.3)}{\leq} (1 + \alpha)(1 - p) \mathbb{E} \left\| \Theta^{(t-1)} - \overline{\Theta}^{(t-1)} \right\|_F^2 + (1 + \alpha^{-1}) \eta_{t-1}^2 T_3.
\end{aligned}
$$

We now bound $T_3$ by relying on Assumption 4:

$$
\begin{aligned}
T_3 = \mathbb{E} \Bigg\| &\left( \nabla F(\Theta^{(t-1)}, Z^{(t-1)}) - \nabla F(\overline{\Theta}^{(t-1)}, Z^{(t-1)}) + \nabla F(\overline{\Theta}^{(t-1)}, Z^{(t-1)}) \right) \cdot \\
&\qquad\qquad\qquad\qquad\qquad\qquad \cdot \left( W^{(t-1)\mathsf{T}} - \frac{\mathbf{1}\mathbf{1}^\mathsf{T}}{n} \right) \Bigg\|_F^2 \\
\overset{(17)}{\leq} 2\mathbb{E} &\left\| \left( \nabla F(\Theta^{(t-1)}, Z^{(t-1)}) - \nabla F(\overline{\Theta}^{(t-1)}, Z^{(t-1)}) \right) \left( W^{(t-1)\mathsf{T}} - \frac{\mathbf{1}\mathbf{1}^\mathsf{T}}{n} \right) \right\|_F^2 \\
&\qquad + 2\mathbb{E} \left\| \nabla F(\overline{\Theta}^{(t-1)}, Z^{(t-1)}) \left( W^{(t-1)\mathsf{T}} - \frac{\mathbf{1}\mathbf{1}^\mathsf{T}}{n} \right) \right\|_F^2 \\
= 2\mathbb{E} &\left\| \left( \nabla F(\Theta^{(t-1)}, Z^{(t-1)}) - \nabla F(\overline{\Theta}^{(t-1)}, Z^{(t-1)}) \right) \left( W^{(t-1)\mathsf{T}} - \frac{\mathbf{1}\mathbf{1}^\mathsf{T}}{n} \right) \right\|_F^2 \\
&\quad + 2 \sum_{i=1}^{n} \mathbb{E} \left\| \sum_{j=1}^{n} W_{ij}^{(t-1)} \nabla F_j(\bar{\theta}^{(t-1)}, Z_j^{(t-1)}) - \frac{1}{n} \sum_{j=1}^{n} \nabla F_j(\bar{\theta}^{(t-1)}, Z_j^{(t-1)}) \right\|_2^2 \\
\overset{(3)}{\leq} 2\mathbb{E} &\left\| \left( \nabla F(\Theta^{(t-1)}, Z^{(t-1)}) - \nabla F(\overline{\Theta}^{(t-1)}, Z^{(t-1)}) \right) \left( W^{(t-1)\mathsf{T}} - \frac{\mathbf{1}\mathbf{1}^\mathsf{T}}{n} \right) \right\|_F^2 + 2n\bar{\tau}^2.
\end{aligned}
$$

For conciseness, we will denote $F_i(\theta_i^{(t-1)}, Z_j^{(t-1)})$ by $F_i(\theta_i^{(t-1)})$ and $\nabla F(\Theta, Z^{(t-1)})$ by $\nabla F(\Theta)$. Using Assumption 3, we can bound the first term of the previous equation by:

$$2(1-p)\mathbb{E}\left\|\left(\nabla F(\Theta^{(t-1)}) - \nabla F(\overline{\Theta}^{(t-1)})\right) - \left(\overline{\nabla F(\Theta^{(t-1)}) - \nabla F(\overline{\Theta}^{(t-1)})}\right)\right\|_F^2$$

$$\overset{(17)}{\leq} 4(1-p)\left[\mathbb{E}\left\|\nabla F(\Theta^{(t-1)}) - \nabla F(\overline{\Theta}^{(t-1)})\right\|_F^2 + \mathbb{E}\left\|\overline{\nabla F(\Theta^{(t-1)}) - \nabla F(\overline{\Theta}^{(t-1)})}\right\|_F^2\right]$$

$$= 4(1-p)\times$$

$$\times \left[\sum_{i=1}^n \left(\mathbb{E}\left\|\nabla F_i(\theta_i^{(t-1)}) - \nabla F_i(\bar{\theta}^{(t-1)})\right\|_2^2 + \mathbb{E}\left\|\frac{1}{n}\sum_{j=1}^n\left(\nabla F_j(\theta_j^{(t-1)}) - \nabla F_j(\bar{\theta}^{(t-1)})\right)\right\|_2^2\right)\right]$$

$$\overset{(A.1)}{\leq} 4(1-p)\left[L^2\sum_{i=1}^n \mathbb{E}\left\|\theta_i^{(t-1)} - \bar{\theta}^{(t-1)}\right\|_2^2 + \frac{n}{n^2}\mathbb{E}\left\|\sum_{j=1}^n\left(\nabla F_j(\theta_j^{(t-1)}) - \nabla F_j(\bar{\theta}^{(t-1)})\right)\right\|_2^2\right]$$

$$\overset{(17)}{\leq} 4(1-p)\left[L^2\sum_{i=1}^n \mathbb{E}\left\|\theta_i^{(t-1)} - \bar{\theta}^{(t-1)}\right\|_2^2 + \sum_{j=1}^n\mathbb{E}\left\|\nabla F_j(\theta_j^{(t-1)}) - \nabla F_j(\bar{\theta}^{(t-1)})\right\|_2^2\right]$$

$$\overset{(A.1)}{\leq} 4(1-p)\left[L^2\sum_{i=1}^n \mathbb{E}\left\|\theta_i^{(t-1)} - \bar{\theta}^{(t-1)}\right\|_2^2 + L^2\sum_{j=1}^n\mathbb{E}\left\|\theta_j^{(t-1)} - \bar{\theta}^{(t-1)}\right\|_2^2\right]$$

$$= 8(1-p)L^2\mathbb{E}\left\|\Theta^{(t-1)} - \overline{\Theta}^{(t-1)}\right\|_F^2.$$

Combining all previous results and setting $\alpha = \frac{p}{2}$, we get:

$$\mathbb{E}\left\|\Theta^{(t)} - \overline{\Theta}^{(t)}\right\|_F^2 \leq (1+\alpha)(1-p)\mathbb{E}\left\|\Theta^{(t-1)} - \overline{\Theta}^{(t-1)}\right\|_F^2$$

$$+ 8(1+\alpha^{-1})(1-p)L^2\eta_{t-1}^2\mathbb{E}\left\|\Theta^{(t-1)} - \overline{\Theta}^{(t-1)}\right\|_F^2 + 2(1+\alpha^{-1})\eta_{t-1}^2 n\bar{\tau}^2$$

$$\leq \underbrace{\left(1+\frac{p}{2}\right)(1-p)}_{\leq 1-\frac{p}{2}}\mathbb{E}\left\|\Theta^{(t-1)} - \overline{\Theta}^{(t-1)}\right\|_F^2$$

$$+ \underbrace{8\left(1+\frac{2}{p}\right)(1-p)}_{\leq \frac{16}{p}}L^2\eta_{t-1}^2\mathbb{E}\left\|\Theta^{(t-1)} - \overline{\Theta}^{(t-1)}\right\|_F^2 + \underbrace{2\left(1+\frac{2}{p}\right)}_{\leq \frac{6}{p}}\eta_{t-1}^2 n\bar{\tau}^2.$$

Since by hypothesis we have $\eta_{t-1} \leq \frac{p}{8L}$, we can bound the second term and get:

$$\mathbb{E}\left\|\Theta^{(t)} - \overline{\Theta}^{(t)}\right\|_F^2 \leq \left(1 - \frac{p}{2} + \frac{p}{4}\right)\mathbb{E}\left\|\Theta^{(t-1)} - \overline{\Theta}^{(t-1)}\right\|_F^2 + \frac{6n\bar{\tau}^2}{p}\eta_{t-1}^2$$

$$= \left(1 - \frac{p}{4}\right)\mathbb{E}\left\|\Theta^{(t-1)} - \overline{\Theta}^{(t-1)}\right\|_F^2 + \frac{6n\bar{\tau}^2}{p}\eta_{t-1}^2.$$

$\square$

**Corollary 1** (Consensus recursion). *Consider the setting of Theorem 1 and fix $\eta_t = \eta \leq \frac{p}{8L}$, we have:*

$$\frac{1}{T+1}\sum_{t=0}^T \mathbb{E}\left\|\Theta^{(t)} - \overline{\Theta}^{(t)}\right\|_F^2 \leq \frac{24\eta^2 n\bar{\tau}^2}{p^2}. \tag{23}$$

*Proof.* Unrolling the expression (22) in Lemma 3 up to $t = 0$, we have for all $t > 0$:

$$\mathbb{E}\left\|\Theta^{(t)} - \overline{\Theta}^{(t)}\right\|_F^2 \le \left(1 - \frac{p}{4}\right)^t \underbrace{\left\|\Theta^{(0)} - \overline{\Theta}^{(0)}\right\|_F^2}_{=0} + \frac{6n\bar{\tau}^2}{p}\eta^2 \sum_{j=0}^{t-1}\left(1 - \frac{p}{4}\right)^j$$

$$= \frac{6n\bar{\tau}^2}{p}\eta^2 \times \frac{1 - \left(1 - \frac{p}{4}\right)^t}{1 - \left(1 - \frac{p}{4}\right)}$$

$$\le \frac{6\eta^2 n\bar{\tau}^2}{p} \times \frac{4}{p}$$

$$= \frac{24\eta^2 n\bar{\tau}^2}{p^2}$$

Summing and dividing by $T + 1$, we get the final result. $\qquad\square$

**Lemma 4** (Convergence rate with $T$ - Convex case). *Consider the setting of Theorem 1 in the convex case. There exists a constant stepsize $\eta \le \eta_{\max} = \frac{p}{8L}$ such that*

$$\frac{1}{T+1}\sum_{t=0}^{T}\mathbb{E}(f(\bar{\theta}^{(t)}) - f^\star) \le 2\left(\frac{br_0}{T+1}\right)^{\frac{1}{2}} + 2e^{\frac{1}{3}}\left(\frac{r_0}{T+1}\right)^{\frac{2}{3}} + \frac{dr_0}{T+1}, \tag{24}$$

*where $b = \frac{\bar{\sigma}^2}{n}$, $e = \frac{36L\bar{\tau}^2}{p^2}$, $d = \frac{8L}{p}$ and $r_0 = \|\theta^{(0)} - \theta^\star\|_2^2$.*

*Proof.* Thanks to the descent lemma (Lemma 1), we almost surely have:

$$f(\bar{\theta}^{(t)}) - f^\star \le \frac{1}{\eta}\left(\left\|\bar{\theta}^{(t)} - \theta^\star\right\|^2 - \mathbb{E}_{Z^{(t)}|\mathcal{F}_{t-1}}\left\|\bar{\theta}^{(t+1)} - \theta^\star\right\|^2 + \frac{\eta^2\bar{\sigma}^2}{n} + \frac{3L}{2n}\eta\left\|\Theta^{(t)} - \overline{\Theta}^{(t)}\right\|_F^2\right),$$

where all terms are $\mathcal{F}_{t-1}$-measurable. Therefore,

$$\mathbb{E}(f(\bar{\theta}^{(t)}) - f^\star) \le \frac{1}{\eta}\left(\mathbb{E}\left\|\bar{\theta}^{(t)} - \theta^\star\right\|^2 - \mathbb{E}\left\|\bar{\theta}^{(t+1)} - \theta^\star\right\|^2 + \frac{\eta^2\bar{\sigma}^2}{n} + \frac{3L}{2n}\eta\mathbb{E}\left\|\Theta^{(t)} - \overline{\Theta}^{(t)}\right\|_F^2\right),$$

and summing up we get:

$$\frac{1}{T+1}\sum_{t=0}^{T}\mathbb{E}(f(\bar{\theta}^{(t)}) - f^\star)$$

$$\le \frac{1}{\eta(T+1)}\sum_{t=0}^{T}\left(\mathbb{E}\left\|\bar{\theta}^{(t)} - \theta^\star\right\|^2 - \mathbb{E}\left\|\bar{\theta}^{(t+1)} - \theta^\star\right\|^2 + \frac{\eta^2\bar{\sigma}^2}{n} + \frac{3L}{2n}\eta\mathbb{E}\left\|\Theta^{(t)} - \overline{\Theta}^{(t)}\right\|_F^2\right)$$

$$\le \frac{1}{\eta(T+1)}\left\|\theta^{(0)} - \theta^\star\right\|^2 + \frac{\eta\bar{\sigma}^2}{n} + \frac{3L}{2n}\frac{1}{T+1}\sum_{t=0}^{T}\mathbb{E}\left\|\Theta^{(t)} - \overline{\Theta}^{(t)}\right\|_F^2$$

$$\overset{(23)}{\le} \frac{1}{\eta(T+1)}\left\|\theta^{(0)} - \theta^\star\right\|^2 + \frac{\bar{\sigma}^2}{n}\eta + \frac{36L\bar{\tau}^2}{p^2}\eta^2.$$

Fixing $\eta = \min\left\{\left(\frac{r_0}{b(T+1)}\right)^{\frac{1}{2}}, \left(\frac{r_0}{e(T+1)}\right)^{\frac{1}{3}}, \frac{1}{d}\right\}$ with $b = \frac{\bar{\sigma}^2}{n}$, $e = \frac{36L\bar{\tau}^2}{p^2}$, $d = \frac{8L}{p}$ and $r_0 = \|\theta^{(0)} - \theta^\star\|_2^2$, then applying Lemma 6 that is recalled after, we obtain the final result. $\qquad\square$

**Lemma 5** (Convergence rate with $T$ - Non convex case). *Consider the setting of Theorem 1 in the non-convex case. There exists a constant stepsize $\eta \le \eta_{\max} = \frac{p}{8L}$ such that*

$$\frac{1}{T+1}\sum_{t=0}^{T}\mathbb{E}\left\|\nabla f(\bar{\theta}^{(t)})\right\|_2^2 \le 2\left(\frac{4bf_0}{T+1}\right)^{\frac{1}{2}} + 2e^{\frac{1}{3}}\left(\frac{4f_0}{T+1}\right)^{\frac{2}{3}} + \frac{4df_0}{T+1}, \tag{25}$$

*where $b = \frac{2L\bar{\sigma}^2}{n}$, $e = \frac{96L^2\bar{\tau}^2}{p^2}$, $d = \frac{8L}{p}$ and $f_0 = f(\theta^{(0)}) - f^\star$.*

*Proof.* Similarly to Lemma 4 for the convex case, we can use the descent Lemma 2 and obtain

$$\mathbb{E}\left\|\nabla f(\bar{\theta}^{(t)})\right\|_2^2 \leq \frac{4}{\eta}\left(\mathbb{E}f_t - \mathbb{E}f_{t+1} + \frac{L^2\eta}{n}\mathbb{E}\left\|\Theta^{(t)} - \overline{\Theta}^{(t)}\right\|_F^2 + \frac{L\bar{\sigma}^2}{2n}\eta^2\right),$$

where for all $t \geq 0$, $f_t \triangleq f(\bar{\theta}^{(t)}) - f^\star$. Then summing up and dividing by $T+1$ we get:

$$\frac{1}{T+1}\sum_{t=0}^{T}\mathbb{E}\left\|\nabla f(\bar{\theta}^{(t)})\right\|_2^2 \leq \frac{4f_0}{\eta(T+1)} + \frac{4L^2}{n}\frac{1}{T+1}\sum_{t=0}^{T}\mathbb{E}\left\|\Theta^{(t)} - \overline{\Theta}^{(t)}\right\|_F^2 + \frac{2L\bar{\sigma}^2}{n}\eta$$

$$\overset{(23)}{\leq} \frac{4f_0}{\eta(T+1)} + \frac{4L^2}{n}\frac{24\eta^2 n\bar{\tau}^2}{p^2} + \frac{2L\bar{\sigma}^2}{n}\eta$$

$$= \frac{4f_0}{\eta(T+1)} + \frac{96L^2\bar{\tau}^2}{p^2}\eta^2 + \frac{2L\bar{\sigma}^2}{n}\eta\,.$$

Fixing $\eta = \min\left\{\left(\frac{4f_0}{b(T+1)}\right)^{\frac{1}{2}}, \left(\frac{4f_0}{e(T+1)}\right)^{\frac{1}{3}}, \frac{1}{d}\right\}$ with $b = \frac{2L\bar{\sigma}^2}{n}$, $e = \frac{96L^2\bar{\tau}^2}{p^2}$, $d = \frac{8L}{p}$ and $f_0 = f(\theta^{(0)}) - f^\star$, we can apply Lemma 6 and obtain the final result. $\square$

**Lemma 6** (Tuning stepsize [17]). *For any parameter $r_0, b, e, d \geq 0$, $T \in \mathbb{N}$, we can fix*

$$\eta = \min\left\{\left(\frac{r_0}{b(T+1)}\right)^{\frac{1}{2}}, \left(\frac{r_0}{e(T+1)}\right)^{\frac{1}{3}}, \frac{1}{d}\right\} \leq \frac{1}{d},$$

*and get*

$$\frac{r_0}{\eta(T+1)} + b\eta + e\eta^2 \leq 2\left(\frac{br_0}{T+1}\right)^{\frac{1}{2}} + 2e^{\frac{1}{3}}\left(\frac{r_0}{T+1}\right)^{\frac{2}{3}} + \frac{dr_0}{T+1}.$$

*Proof.* The proof of this lemma can be found in the supplementary materials of [17] (Lemma 15). $\square$

# D  Additional Results and Proofs

**Proposition 1.** *Let Assumptions 2-3 and 5 to be verified. Then Assumption 4 is satisfied with $\bar{\tau}^2 = (1-p)\left(\bar{\zeta}^2 + \bar{\sigma}^2\right)$, where $\bar{\sigma}^2 \triangleq \frac{1}{n}\sum_i \sigma_i^2$.*

*Proof.* Denoting $\nabla F(\theta) = (\nabla F_1(\theta, Z_1), \ldots, \nabla F_n(\theta, Z_n)) \in \mathbb{R}^{d\times n}$, and using the relation

$$\mathbb{E}\|Y\|_2^2 = \|\mathbb{E}Y\|_2^2 + \mathbb{E}\|Y - \mathbb{E}Y\|_2^2, \tag{26}$$

we have:

$$H^{(t)} = \frac{1}{n}\mathbb{E}\|\nabla F(\theta)W^{(t)} - \overline{\nabla F(\theta)}\|_F^2$$

$$\overset{(A.3)}{\leq} \frac{1-p}{n}\mathbb{E}\|\nabla F(\theta) - \overline{\nabla F(\theta)}\|_F^2$$

$$= \frac{1-p}{n}\sum_{i=1}^{n}\mathbb{E}\left\|\nabla F_i(\theta, Z_i) - \frac{1}{n}\sum_{j=1}^{n}\nabla F_j(\theta, Z_j)\right\|_2^2$$

$$\overset{(26)}{=} \frac{1-p}{n}\sum_{i=1}^{n}\left(\left\|\nabla f_i(\theta) - \frac{1}{n}\sum_{j=1}^{n}\nabla f_j(\theta)\right\|_2^2 + \mathbb{E}\left\|\sum_{j=1}^{n}\left(\mathbb{1}_{\{j=i\}} - \frac{1}{n}\right)(\nabla F_j(\theta, Z_j) - \nabla f_j(\theta))\right\|_2^2\right)$$

$$\overset{(A.5)}{\leq} (1-p)\left(\bar{\zeta}^2 + \frac{1}{n}\sum_{i=1}^{n}\mathbb{E}\left\|\sum_{j=1}^{n}\left(\mathbb{1}_{\{j=i\}} - \frac{1}{n}\right)(\nabla F_j(\theta, Z_j) - \nabla f_j(\theta))\right\|_2^2\right).$$

Since all terms $j$ in the norm are independent and with expectation $0$, the expectation of the sum is equal to the sum of expectations and

$$
\begin{aligned}
H^{(t)} &\leq (1-p)\left(\bar{\zeta}^2 + \frac{1}{n}\sum_{i=1}^{n}\sum_{j=1}^{n}\left(\mathbb{1}_{\{j=i\}} - \frac{1}{n}\right)^2 \mathbb{E}\left\|\nabla F_j(\theta, Z_j) - \nabla f_j(\theta)\right\|_2^2\right) \\
&= (1-p)\left(\bar{\zeta}^2 + \frac{1}{n}\sum_{j=1}^{n}\mathbb{E}\left\|\nabla F_j(\theta, Z_j) - \nabla f_j(\theta)\right\|_2^2 \underbrace{\sum_{i=1}^{n}\left(\mathbb{1}_{\{j=i\}} - \frac{1}{n}\right)^2}_{=\frac{n-1}{n}}\right) \\
&\overset{(A.2)}{\leq} (1-p)\left(\bar{\zeta}^2 + \frac{n-1}{n}\bar{\sigma}^2\right) \leq (1-p)\left(\bar{\zeta}^2 + \bar{\sigma}^2\right),
\end{aligned}
$$

which concludes the proof. $\qquad\square$

**Proposition 2.** (Upper-bound on $H$ under label skew) *Consider the statistical framework with label-skew and assume there exists $B > 0$ such that $\forall k = 1,\ldots,K$ and $\forall \theta \in \mathbb{R}^d$, $\|\mathbb{E}_X[\nabla F(\theta; X, Y)|Y = k] - \frac{1}{K}\sum_{k'=1}^{K}\mathbb{E}_X[\nabla F(\theta; X, Y)|Y = k']\|_2^2 \leq B$. Then, denoting $\pi_{jk} \triangleq P_j(Y = k)$, we can bound the neighborhood heterogeneity $H$ in Equation (3) by:*

$$
H \leq \frac{KB}{n}\sum_{k=1}^{K}\sum_{i=1}^{n}\left(\sum_{j=1}^{n}W_{ij}\pi_{jk} - \frac{1}{n}\sum_{j=1}^{n}\pi_{jk}\right)^2 + \frac{\sigma_{\max}^2}{n}\left\|W - \frac{1}{n}\mathbf{1}\mathbf{1}^\mathsf{T}\right\|_F^2 .
$$

*Proof.* First, observe that the local objective functions can be re-written

$$
\begin{aligned}
f_j(\theta) &= \mathbb{E}_{(X,Y)\sim\mathcal{D}_j}[F(\theta; X, Y)] \\
&= \sum_{k=1}^{K}P_j(Y = k)\mathbb{E}_X[F(\theta; X, Y)|Y = k] \\
&= \sum_{k=1}^{K}\pi_{jk}\mathbb{E}_X[F(\theta; X, Y)|Y = k] .
\end{aligned}
$$

From (4), we have the bias-variance decomposition

$$
\begin{aligned}
H &\leq \frac{1}{n}\sum_{i=1}^{n}\left\|\sum_{j=1}^{n}W_{ij}\nabla f_j(\theta) - \nabla f(\theta)\right\|_2^2 + \frac{\sigma_{\max}^2}{n}\left\|W - \frac{1}{n}\mathbf{1}\mathbf{1}^\mathsf{T}\right\|_F^2 \\
&= \frac{1}{n}\sum_{i=1}^{n}\left\|\sum_{j=1}^{n}(W_{ij} - \frac{1}{n})\nabla f_j(\theta)\right\|_2^2 + \frac{\sigma_{\max}^2}{n}\left\|W - \frac{1}{n}\mathbf{1}\mathbf{1}^\mathsf{T}\right\|_F^2 \\
&= \frac{1}{n}\sum_{i=1}^{n}\underbrace{\left\|\sum_{j=1}^{n}(W_{ij} - \frac{1}{n})\sum_{k=1}^{K}\pi_{jk}\mathbb{E}_X[\nabla F(\theta; X, Y)|Y = k]\right\|_2^2}_{T_4} + \frac{\sigma_{\max}^2}{n}\left\|W - \frac{1}{n}\mathbf{1}\mathbf{1}^\mathsf{T}\right\|_F^2 .
\end{aligned}
$$

Then, observing that $\sum_{j=1}^{n}(W_{ij} - \frac{1}{n}) = 0$ and $\sum_{k=1}^{K}\pi_{jk} = 1$ imply

$$\sum_{j=1}^{n}(W_{ij} - \frac{1}{n}) \sum_{k=1}^{K} \pi_{jk} \frac{1}{K} \sum_{k'=1}^{n} \mathbb{E}_X[\nabla F(\theta; X, Y)|Y = k'] = \mathbf{0} \,,$$

we can add this in the norm of the term $T_4$ defined above and get

$$T_4 = \left\| \sum_{j=1}^{n}(W_{ij} - \frac{1}{n}) \sum_{k=1}^{K} \pi_{jk} \Big( \mathbb{E}_X[\nabla F(\theta; X, Y)|Y = k] - \frac{1}{K} \sum_{k'=1}^{K} \mathbb{E}_X[\nabla F(\theta; X, Y)|Y = k'] \Big) \right\|_2^2$$

$$= \left\| \sum_{k=1}^{K} \Big( \mathbb{E}_X[\nabla F(\theta; X, Y)|Y = k] - \frac{1}{K} \sum_{k'=1}^{K} \mathbb{E}_X[\nabla F(\theta; X, Y)|Y = k'] \Big) \sum_{j=1}^{n}(W_{ij} - \frac{1}{n})\pi_{jk} \right\|_2^2$$

$$\overset{(17)}{\leq} K \sum_{k=1}^{K} \left\| \Big( \mathbb{E}_X[\nabla F(\theta; X, Y)|Y = k] - \frac{1}{K} \sum_{k'=1}^{K} \mathbb{E}_X[\nabla F(\theta; X, Y)|Y = k'] \Big) \sum_{j=1}^{n}(W_{ij} - \frac{1}{n})\pi_{jk} \right\|_2^2$$

$$= K \sum_{k=1}^{K} \underbrace{\left\| \mathbb{E}_X[\nabla F(\theta; X, Y)|Y = k] - \frac{1}{K} \sum_{k'=1}^{K} \mathbb{E}_X[\nabla F(\theta; X, Y)|Y = k'] \right\|_2^2}_{\leq B}$$

$$\times \Big( \sum_{j=1}^{n}(W_{ij} - \frac{1}{n})\pi_{jk} \Big)^2$$

$$\leq KB \sum_{k=1}^{K} \Big( \sum_{j=1}^{n} W_{ij}\pi_{jk} - \frac{1}{n} \sum_{j=1}^{n} \pi_{jk} \Big)^2 \,.$$

Finally, plugging this into the upper-bound on $H$ found above, we get the final result. $\qquad \square$

**Theorem 2** *Consider the statistical setup presented in Section 4.1 and let $\{\widehat{W}^{(l)}\}_{l=1}^{L}$ be the sequence of mixing matrices generated by Algorithm 2. Then, at any iteration $l = 1, \dots, L$, we have:*

$$g(\widehat{W}^{(l)}) \leq \tfrac{16}{l+2} \big( \lambda + \tfrac{1}{n} \big\| \sum_{k=1}^{K} (\Pi_{:,k} - \overline{\Pi_{:,k}} \mathbf{1}) \cdot \Pi_{:,k}^{\mathsf{T}} \big\|_2^\star \big) \,,$$

*where $\|\cdot\|_2^\star$ stands for the nuclear norm, i.e., the sum of singular values. Bounding the second term in the parenthesis, we can obtain the looser bound*

$$g(\widehat{W}^{(l)}) \leq \tfrac{16}{l+2} (\lambda + 1) \,.$$

*Furthermore, we have $d_{max}^{in}(\widehat{W}^{(l)}) \leq l$ and $d_{max}^{out}(\widehat{W}^{(l)}) \leq l$, resulting in a per-iteration complexity bounded by $l$.*

*Proof.* The proof of this theorem is directly derived from Theorem 3 given below, applied with the parameters of our problem. To prove the first inequality, we first need to find a bound on the diameter of the set of doubly stochastic matrices, denoted $\mathrm{diam}_{\|\cdot\|}(\mathcal{S})$, for a certain (matrix) norm $\|\cdot\|$. We fix this norm to be the operator norm induced by the $\ell_2$-norm, denoted $\|\cdot\|_2$, which is simply the maximum singular value of the matrix.

For all $W, P \in \mathcal{S}$, we have

$$\|W - P\|_2 \leq \|W\|_2 + \|P\|_2$$
$$= 1 + 1 = 2 \,,$$

which comes from the fact that $W$ and $P$ are doubly stochastic, i.e., their largest eigenvalue is 1. This shows that $\mathrm{diam}_{\|\cdot\|_2}(\mathcal{S}) \leq 2$.

Let us now find the Lipschitz constant associated to the gradient of the objective:

$$\nabla g(W) = \frac{2}{n} \sum_{k=1}^{K} (W\Pi_{:,k} - \overline{\Pi_{:,k}}\mathbf{1}) \cdot \Pi_{:,k}^{\mathsf{T}} + \frac{2}{n}\lambda \left( W - \frac{\mathbf{1}\mathbf{1}^{\mathsf{T}}}{n} \right) .$$

Recall that the dual norm $\|\cdot\|_1^{\star}$ of $\|\cdot\|_1$ is the nuclear norm, i.e., the sum of the singular values. For any $W, P \in \mathcal{S}$, we have

$$\|\nabla g(W) - \nabla g(P)\|_2^{\star} = \frac{2}{n} \left\| (W - P) \left( \lambda I + \sum_{k=1}^{K} \Pi_{:,k}\Pi_{:,k}^{\mathsf{T}} \right) \right\|_2^{\star}$$

$$\leq \frac{2}{n} \|\lambda(W-P)I\|_2^{\star} + \frac{2}{n} \left\| (W - P) \sum_{k=1}^{K} \Pi_{:,k}\Pi_{:,k}^{\mathsf{T}} \right\|_2^{\star}$$

$$\leq \frac{2\lambda}{n} \|W - P\|_2 \|I\|_2^{\star} + \frac{2}{n} \left\| (W - P) \sum_{k=1}^{K} \Pi_{:,k}\Pi_{:,k}^{\mathsf{T}} \right\|_2^{\star},$$

where the last inequality is obtained using the fact that for any real matrices $A$ and $B$, $\|AB\|^{\star} \leq \|A^{\mathsf{T}}\|\|B\|^{\star}$.

Before bounding the second term, we must observe that because $W$ and $P$ are doubly stochastic, $(W - P)\mathbf{1} = 0$ and therefore, for any matrix $A \in \mathbb{R}^{n \times n}$, $(W - P)A = (W - P)(A - \frac{\mathbf{1}\mathbf{1}^{\mathsf{T}}}{n}A)$.

Now, the second term can be re-written and bounded as follows:

$$\frac{2}{n} \left\| (W - P) \sum_{k=1}^{K} \Pi_{:,k}\Pi_{:,k}^{\mathsf{T}} \right\|_2^{\star} = \frac{2}{n} \left\| (W - P) \left( \sum_{k=1}^{K} \Pi_{:,k}\Pi_{:,k}^{\mathsf{T}} - \frac{\mathbf{1}\mathbf{1}^{\mathsf{T}}}{n} \sum_{k=1}^{K} \Pi_{:,k}\Pi_{:,k}^{\mathsf{T}} \right) \right\|_2^{\star}$$

$$\leq \frac{2}{n} \|W - P\|_2 \left\| \sum_{k=1}^{K} \Pi_{:,k}\Pi_{:,k}^{\mathsf{T}} - \frac{\mathbf{1}\mathbf{1}^{\mathsf{T}}}{n} \sum_{k=1}^{K} \Pi_{:,k}\Pi_{:,k}^{\mathsf{T}} \right\|_2^{\star}$$

$$= \frac{2}{n} \|W - P\|_2 \left\| \sum_{k=1}^{K} (\Pi_{:,k} - \overline{\Pi_{:,k}}\mathbf{1}) \cdot \Pi_{:,k}^{\mathsf{T}} \right\|_2^{\star} .$$

Plugging the previous result into the bound obtained above, and since $\|I\|_2^{\star} = n$, we get

$$\|\nabla g(W) - \nabla g(P)\|_2^{\star} \leq 2 \left( \lambda + \frac{1}{n} \left\| \sum_{k=1}^{K} (\Pi_{:,k} - \overline{\Pi_{:,k}}\mathbf{1}) \cdot \Pi_{:,k}^{\mathsf{T}} \right\|_2^{\star} \right) \|W - P\|_2 .$$

We can now apply Theorem 3 with the found Lipschitz constant and diameter, which gives:

$$g(\widehat{W}^{(l)}) - g(W^{\star}) \leq \frac{16}{l+2} \left( \lambda + \frac{1}{n} \left\| \sum_{k=1}^{K} (\Pi_{:,k} - \overline{\Pi_{:,k}}\mathbf{1}) \cdot \Pi_{:,k}^{\mathsf{T}} \right\|_2^{\star} \right) ,$$

where $W^{\star}$ is the optimal solution of the problem. Since we known that $W^{\star} = \frac{\mathbf{1}\mathbf{1}^{\mathsf{T}}}{n}$ with $g(W^{\star}) = 0$, we obtain the first inequality in Theorem 2.

To prove the second inequality, it suffices to show that $\left\|\sum_{k=1}^{K}(\Pi_{:,k} - \overline{\Pi_{:,k}}\mathbf{1}) \cdot \Pi_{:,k}^{\mathsf{T}}\right\|_2^\star \leq n$. We have:

$$\left\|\sum_{k=1}^{K}(\Pi_{:,k} - \overline{\Pi_{:,k}}\mathbf{1}) \cdot \Pi_{:,k}^{\mathsf{T}}\right\|_2^\star = \left\|\left(I - \frac{\mathbf{1}\mathbf{1}^{\mathsf{T}}}{n}\right)\sum_{k=1}^{K}\Pi_{:,k}\Pi_{:,k}^{\mathsf{T}}\right\|_2^\star$$

$$\leq \left\|I - \frac{\mathbf{1}\mathbf{1}^{\mathsf{T}}}{n}\right\|_2 \left\|\sum_{k=1}^{K}\Pi_{:,k}\Pi_{:,k}^{\mathsf{T}}\right\|_2^\star$$

$$= \left\|\sum_{k=1}^{K}\Pi_{:,k}\Pi_{:,k}^{\mathsf{T}}\right\|_2^\star$$

$$\leq \sum_{k=1}^{K}\left\|\Pi_{:,k}\Pi_{:,k}^{\mathsf{T}}\right\|_2^\star .$$

Because for any $k = 1, \ldots, K$, $\Pi_{:,k}\Pi_{:,k}^{\mathsf{T}}$ is a rank-1 matrix, its unique eigenvalue is $\Pi_{:,k}^{\mathsf{T}}\Pi_{:,k}$ and therefore

$$\left\|\sum_{k=1}^{K}(\Pi_{:,k} - \overline{\Pi_{:,k}}\mathbf{1}) \cdot \Pi_{:,k}^{\mathsf{T}}\right\|_2^\star \leq \sum_{k=1}^{K}\left\|\Pi_{:,k}\Pi_{:,k}^{\mathsf{T}}\right\|_2^\star$$

$$= \sum_{k=1}^{K}\Pi_{:,k}^{\mathsf{T}}\Pi_{:,k}$$

$$= \sum_{k=1}^{K}\sum_{i=1}^{n}\pi_{ik}^2$$

$$\overset{\text{Holder}}{\leq} \sum_{i=1}^{n}\max_{k}\{\pi_{ik}\}\underbrace{\sum_{k=1}^{K}\pi_{ik}}_{=1}$$

$$\leq \sum_{i=1}^{n}1 = n ,$$

which concludes the proof of the second inequality in Theorem 2.

The last statement of the theorem follows directly from the structure of permutation matrices and the greedy nature of the algorithm. $\qquad\square$

**Theorem 3.** (Frank-Wolfe Convergence [15, 4]) *Let the gradient of the objective function $g : x \rightarrow g(x)$ be L-smooth with respect to a norm $\|\cdot\|$ and its dual norm $\|\cdot\|^\star$:*

$$\|\nabla g(x) - \nabla g(y)\|^\star \leq L\|x - y\| .$$

*If $g$ is minimized over $\mathcal{S}$ using Frank-Wolfe algorithm, then for each $l \geq 1$, the iterates $x^{(l)}$ satisfy*

$$g(x^{(l)}) - g(x^\star) \leq \frac{2Ldiam_{\|\cdot\|}(\mathcal{S})^2}{l+2} ,$$

*where $x^\star \in \mathcal{S}$ is an optimal solution of the problem and $diam_{\|\cdot\|}(\mathcal{S})$ stands for the diameter of $\mathcal{S}$ with respect to the norm $\|\cdot\|$.*

*Proof.* The proof of this theorem is a direct combination of Theorem 1 and Lemma 7 in [15], both proved in the paper. □

**Proposition 3.** (Relation between $p$ and $\|W - \frac{\mathbf{1}\mathbf{1}^\mathsf{T}}{n}\|_F^2$) *Let $W$ be a mixing matrix satisfying Assumption 3. Then,*

$$(1 - p) \leq \left\| W - \frac{\mathbf{1}\mathbf{1}^\mathsf{T}}{n} \right\|_F^2 \leq (n - 1)(1 - p) .$$

*Proof.* The upper-bound is a direct application of Assumption 3 with $M = I$, the identity matrix of size $n$:

$$\left\| W^\mathsf{T} - \frac{\mathbf{1}\mathbf{1}^\mathsf{T}}{n} \right\|_F^2 = \left\| IW^\mathsf{T} - I\frac{\mathbf{1}\mathbf{1}^\mathsf{T}}{n} \right\|_F^2 \overset{(A.3)}{\leq} (1 - p) \left\| I - \frac{\mathbf{1}\mathbf{1}^\mathsf{T}}{n} \right\|_F^2 = (1 - p)(n - 1) .$$

To show the lower-bound, denote by $s_1(M), \ldots, s_n(M)$ the (decreasing) singular values of any square matrix $M \in \mathbb{R}^{n \times n}$. Denote similarly $\lambda_1(M), \ldots, \lambda_n(M)$ the eigenvalues of any *symmetric* square matrix $M \in \mathbb{R}^{n \times n}$.

$$\begin{aligned}
\left\| W - \frac{\mathbf{1}\mathbf{1}^\mathsf{T}}{n} \right\|_F^2 = \sum_{i=1}^n s_i^2 \left( W - \frac{\mathbf{1}\mathbf{1}^\mathsf{T}}{n} \right) &\geq s_1^2 \left( W - \frac{\mathbf{1}\mathbf{1}^\mathsf{T}}{n} \right) \\
&= \lambda_1 \left( (W - \frac{\mathbf{1}\mathbf{1}^\mathsf{T}}{n})^\mathsf{T} (W - \frac{\mathbf{1}\mathbf{1}^\mathsf{T}}{n}) \right) \\
&= \lambda_1 \left( W^\mathsf{T} W - \frac{\mathbf{1}\mathbf{1}^\mathsf{T}}{n} \right) \\
&= \lambda_2(W^\mathsf{T} W) \geq 1 - p .
\end{aligned}$$

The last *equality* is obtained by noticing that $W^\mathsf{T} W$ is a symmetric doubly stochastic matrix. It therefore admits an eigenvalue decomposition where the largest eigenvalue 1 is associated with the eigenvector $\frac{1}{\sqrt{n}}\mathbf{1}$. This makes $W^\mathsf{T} W - \frac{\mathbf{1}\mathbf{1}^\mathsf{T}}{n}$ having the eigenvalue 0 associated to the vector $\frac{1}{\sqrt{n}}\mathbf{1}$ and the largest eigenvalue of $W^\mathsf{T} W - \frac{\mathbf{1}\mathbf{1}^\mathsf{T}}{n}$ becomes the second-largest eigenvalue of $W^\mathsf{T} W$. The final *inequality* comes from the fact that Assumption 3 is always true with $p = 1 - \lambda_2(W^\mathsf{T} W)$ which implies that the best $p$ satisfying Assumption 3 in necessarily greater or equal to $1 - \lambda_2(W^\mathsf{T} W)$. □

### D.1 Extension to Random Mixing Matrices

As mentioned in Section 2, all our theoretical results can easily be extended to random mixing matrices. In that framework, at each iteration $t$ of the D-SGD algorithm, the matrix $W^{(t)}$ is sampled from a doubly stochastic matrix distribution denoted $\mathcal{W}^{(t)}$, independent of the iterates of parameters $\theta^{(t)}$, and possibly time-varying.

To obtain the convergence result, we slightly modify Assumption 3 and Assumption 4 by adding an expectation with respect to $W$ in front of the equations. For instance, Assumption 3 becomes $\mathbb{E}_{W \sim \mathcal{W}} \|MW^\mathsf{T} - \overline{M}\|_F^2 \leq (1 - p)\|M - \overline{M}\|_F^2$. Then, the statement of Theorem 1 is also slightly modified by assuming that it is the distributions $\mathcal{W}^{(0)}, \ldots, \mathcal{W}^{(T-1)}$ that must now respect Assumptions 3 and 4.

By appropriately conditioning with respect to the random mixing matrices or with respect to the iterates, the proof of the theorem remains the same.

## E Additional Experiments

### E.1 Simulations on Synthetic Data

**Model.** We generalize the mean estimation objective of the example from section B with $K = 10$ clusters and $n = 100$ nodes, with exactly 10 nodes associated to each cluster. Each cluster is

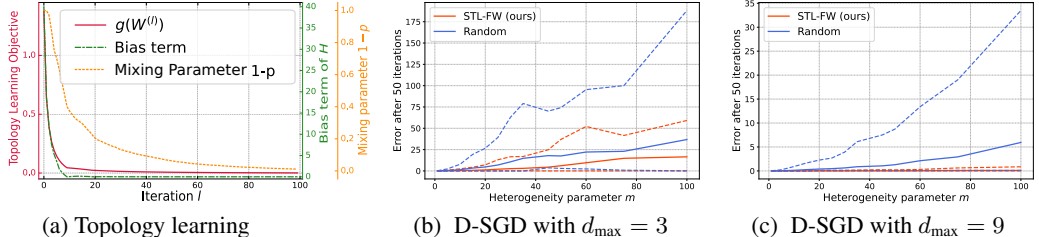

|  |  |  |
|---|---|---|
| (a) Topology learning | (b) D-SGD with $d_{\max} = 3$ | (c) D-SGD with $d_{\max} = 9$ |

**Figure 2: (a)** Evolution of key quantities across the iterations of topology learning: in red the objective function $g(W^{(l)})$, in green the bias term $\frac{1}{n}\sum_{i=1}^{n}\|\sum_{j=1}^{n} W_{ij}^{(l)} \nabla f_j(\theta) - \nabla f(\theta)\|_2^2$ of $H$ and in yellow the mixing parameter $1-p = \lambda_2(W^{(l)\mathsf{T}} W^{(l)})$. Here, $\lambda = 0.5$ and $m = 5$. **(b, c)** Error $n^{-1}\|\theta^{(t)} - \theta^\star\|_2^2$ (solid line) of D-SGD after 50 iterations, averaged over 10 runs, for increasing levels of heterogeneity (measured by parameter $m$). The dashed lines show $\max_i(\theta_i^{(t)} - \theta^\star)^2$ and $\min_i(\theta_i^{(t)} - \theta^\star)^2$, illustrating the variability across nodes.

associated with a Gaussian distribution. The variance of the $K$ distributions is the same ($\tilde{\sigma}^2 = 1$) but their means are evenly spread over $[-m, m]$. Thus, $m \geq 0$ controls the heterogeneity of the problem (the bigger $m$, the more heterogeneous the setup). We can analytically compute all numerical constants introduced throughout the paper. Unless otherwise noted, $\lambda$ is set to $\sigma^2/KB$ where $\sigma^2 = 4\tilde{\sigma}^2$ and $B = 4m^2$. **Competitor.** For a fixed budget $d_{\max}$, we compare the topology learned by STL-FW with a random $d_{\max}$-regular graph having uniform weights $\frac{1}{d_{\max}+1}$. This graph is independent of the data but optimizes the mixing $p$ (every node will have exactly $b$ neighbors, with uniform weights). We use a fixed step-size for D-SGD, which is tuned separately for each topology in the interval $[0.001, 1]$.

**Results.** We first study the behavior of our topology learning algorithm. As seen in Figure 2(a), the objective function $g(W^{(l)})$ decreases quickly in the first iterations with a clear "elbow" at $l = 9$ iterations. This is because we have $K = 10$ classes, hence 9 neighbors are sufficient to compensate for label skew. We also see that decreasing $g$ successfully decreases the two key quantities that affect the convergence of D-SGD and are upper bounded by $g$: the bias term of $H$ (which can be computed exactly in this setup) and the mixing parameter $1 - p$ (which continues to decrease beyond $l = 9$).

Figure 2(b, c) shows that the topology learned by STL-FW indeed translates into faster convergence for D-SGD than with the random (but well-connected) topology in data heterogeneous settings. This is especially striking when looking at best and worst-case errors across nodes (dashed lines). For a low budget ($d_{\max} = 2$), D-SGD with our topology remains slightly impacted by heterogeneity. But remarkably, for $d_{\max} = 9$, our topology makes D-SGD completely insensitive to increasing data heterogeneity. This observation is consistent with the elbow observed at $l = d_{\max} = 9$ in Figure 2(a).

### E.2 Detailed Experimental Setup for the real-world data

Our main goal is to provide a fair comparison of the convergence speed across different topologies in order to show the benefits of the principled approach to topology learning provided by STL-FW. We essentially follow the experimental setup in [2], which we recall below.

In our study, we focus our investigation on the convergence speed, rather than the final accuracy after a fixed number of iterations. Indeed, depending on when training is stopped, the relative difference in final accuracy across different algorithms may vary significantly and lead to different conclusions. Instead of relying on somewhat arbitrary stopping points, we show the convergence curves of generalization performance (i.e., the accuracy on the test set throughout training), up to a point where it is clear that the different approaches have converged, will not make significantly more progress, or behave essentially the same.

**Datasets.** We experiment with two datasets: MNIST [9] and CIFAR10 [20], which both have $K = 10$ classes. For MNIST, we use 50k and 10k examples from the original set for training and testing respectively. For CIFAR10, we used 50k images of the original training set for training and 10k examples of the test set for measuring prediction accuracy.

For both MNIST and CIFAR10, we use the heterogeneous data partitioning scheme proposed by McMahan et al. [27] in their seminal FL work: we sort all training examples by class, then split the list into shards of equal size, and randomly assign two shards to each node. When the number of examples of one class does not divide evenly in shards, as is the case for MNIST, some shards may have examples of more than one class and therefore nodes may have examples of up to 4 classes. However, most nodes will have examples of 2 classes.

**Models.** We use a logistic regression classifier for MNIST, which provides up to 92.5% accuracy in the centralized setting. For CIFAR10, we use a Group-Normalized variant of LeNet [13], a deep convolutional network which achieves an accuracy of $74.15\%$ in the centralized setting. These models are thus reasonably accurate (which is sufficient to study the effect of the topology) while being sufficiently fast to train in a fully decentralized setting and simple enough to configure and analyze. Regarding hyper-parameters, we use the learning rate and mini-batch size found in [2] after cross-validation for $n = 100$ nodes, respectively 0.1 and 128 for MNIST and 0.002 and 20 for CIFAR10.

**Metrics.** We evaluate a network of $n = 100$ nodes, creating multiple models in memory and simulating the exchange of messages between nodes. To ignore the impact of distributed execution strategies and system optimization techniques, we report the test accuracy of all nodes (min, max, average) as a function of the number of times each example of the dataset has been sampled by a node, i.e. an *epoch*. This is equivalent to the classic case of a single node sampling the full distribution. All our results were obtained on a custom version of the *non-iid topology simulator* made available online by the authors of [2],[1] which provides deterministic and fully replicable experiments on top of Pytorch and ensures all topologies were used in the same algorithm implementation and used exactly the same inputs.

**Baselines** We compare our results against an ideal baseline: a fully-connected network topology with the same number of nodes. All other things being equal, any other topology using less edges will converge at the same speed or slower: *this is therefore the most difficult and general baseline to compare against*. This baseline is also essentially equivalent to a centralized (single) IID node using a batch size $n$ times bigger, where $n$ is the number of nodes. Both a fully-connected network and a single IID node effectively optimize a single model and sample uniformly from the global distribution: both thus remove entirely the effect of label distribution skew and of the network topology on the optimization. In practice, we prefer a fully-connected network because it converges slightly faster and obtains slightly better final accuracy than a single node sampling randomly from the global distribution.

We also provide comparisons against popular sparse topologies, such as random graphs and exponential graphs [35]. For the random graph, we use a similar number of edges ($d_{\max}$) per node to determine whether a simple sparse topology could work equally well. For the exponential graph, we follow the deterministic construction of [35] and consider edges to be undirected, resulting in $d_{max} = 14$ for $n = 100$.

We finally compare against D-Cliques [2], the only competitor which takes into account the data heterogeneity in the choice of topology. D-Cliques constructs a topology around sparsely interconnected cliques such that the union of local datasets within a clique is representative of the global distribution, i.e. it minimizes the first term in our objective function (Eq. 8) within each clique.

### E.3 Impact of $\lambda$ in STL-FW

Figure 3 shows the impact of $\lambda$, which rules the bias-variance trade-off in our objective function for learning the topology. We present results for two extreme values, respectively 0.0001 and 1000 as well as middle ground of 0.1. For both datasets, $\lambda$ has little effect on convergence speed. From a practical perspective, this is an advantage as it removes the need for tuning $\lambda$ (one can simply set it to a default positive value). This behavior may be explained by the fact that reducing the bias term alone also leads to a reduction of variance. Hence, the variance term becomes useful only when the bias term has been "erased" (or made very small), which can happen only after a certain number of STL-FW iterations, i.e., for a potentially large $d_{\max}$. For all other experiments, we used $\lambda = 0.1$.

---

[1]`https://gitlab.epfl.ch/sacs/distributed-ml/non-iid-topology-simulator`

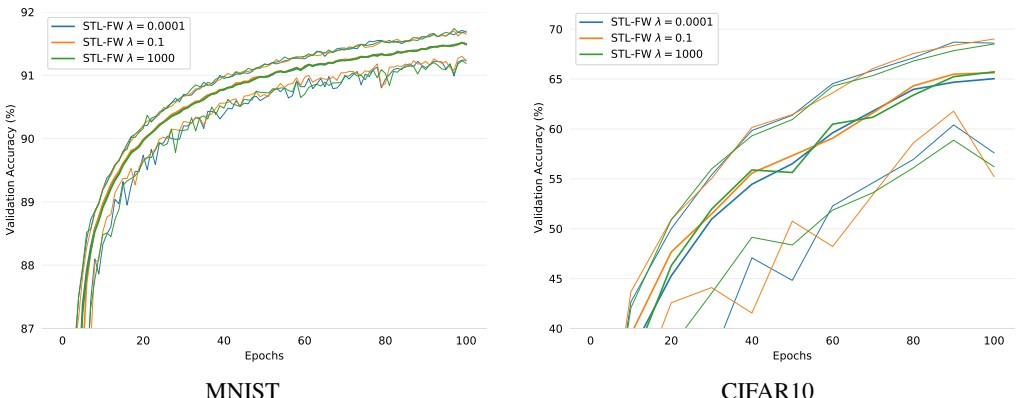

**Figure 3:** Effect of the hyperparameter $\lambda$ of STL-FW on the convergence speed of D-SGD with 100 nodes, $d_{max} = 10$.

### E.4 Impact of $d_{max}$ on STL-FW

Figure 4 shows on a single plot the impact of the communication budget $d_{max}$ of STL-FW on the convergence speed of D-SGD. The communication budget has a strong impact in both cases, with STL-FW providing the same convergence speed as fully-connected when $d_{max} = 99$, but with some residual variance because some nodes end up wth less than 99 edges. Most of the benefits of STL-FW are obtained with the first 10 edges, with additional edges providing only marginal benefits compared to fully-connected. We thus chose to show all experiments of the main text with three budgets, a small $d_{max} = 2$, a medium $d_{max} = 5$, and a large budget $d_{max} = 10$.

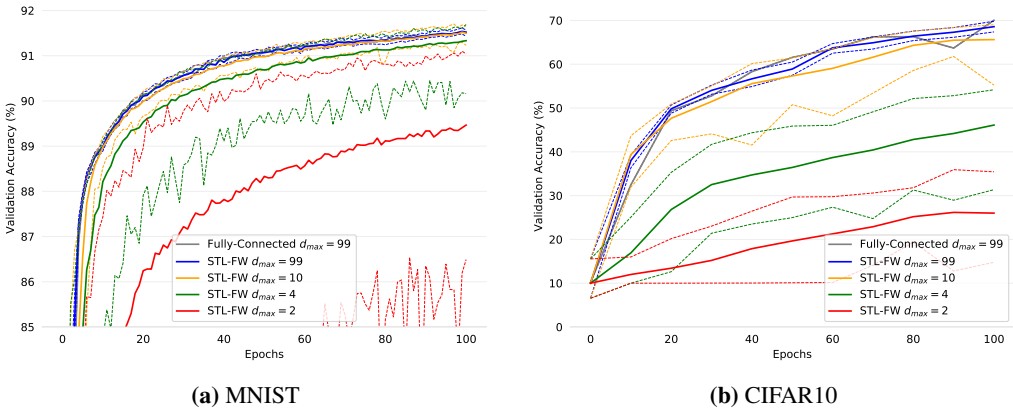

**Figure 4:** Effect of communication budget ($d_{max}$) of STL-FW on the convergence speed of D-SGD with 100 nodes, $\lambda = 0.1$.

Finally, for a small budget $d_{max} = 2$, we had seen in Figure 1 in the main text that STL-FW did not provide significant benefits compared to a random graph on CIFAR10. Figure 5 shows that as soon as $d_{max} = 3$, STL-FW starts providing benefits compared to a random topology on CIFAR10.

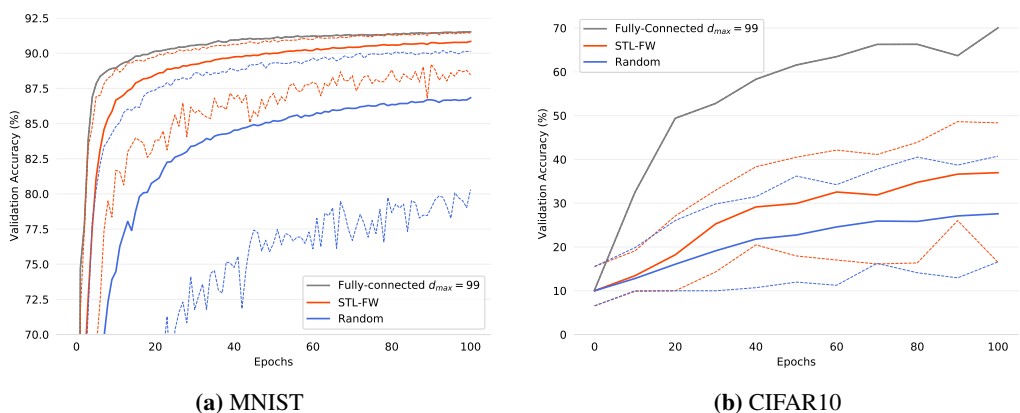

**(a)** MNIST

**(b)** CIFAR10

**Figure 5:** Convergence speed of D-SGD with our STL-FW topology and a random topology under small communication budget $d_{max} = 3$.

