# OpenReview forum: "Refined Convergence and Topology Learning for Decentralized Optimization with Heterogeneous Data"
_NeurIPS.cc/2022/Workshop/Federated_Learning — FL-NeurIPS 2022 Poster_

### Official Review · Reviewer_EKuj · 2022-10-14
**Review for Refined Convergence and Topology Learning for Decentralized Optimization with Heterogeneous Data**

The paper presents refined convergence for D-SGD by considering the connection between topology and data heterogeneity. Specifically, the authors revisit the D-SGD algorithm, develop a new concept called neighborhood heterogeneity, and analyze theoretically the convergence rate. The new concept instead considers the bias-variance tradeoff between local and global gradients, not just only the bias that has been existing in most published works. The authors give a refined rate for the convergence of D-SGD, which reveals the advantage of the neighborhood heterogeneity in the sparse topologies. To control the communication complexity, the authors also present the topology learning technique to minimize the heterogeneity. They use a few simulation and real-world datasets to validate their proposed ideas with a couple simple models.

I think overall, the topic investigated in the paper is interesting. Particularly, data heterogeneity in multi-agent systems is a critical issue that significantly affects the model performance. The presentation in this paper is clear and easy to follow, though the algorithms studied in this work have been well established. However, this makes the overall contributions a bit incremental. Given the current draft, I still have some comments that need to be addressed in the following.

1. What is the significant advantage of this proposed algorithm compared to other existing algorithms except D-SGD? Is this paper just presenting the refined results for only D-SGD?

2. How does this refinement perform when using other bigger datasets and more complex models? Can this refinement be extended to other decentralized algorithms?

3. The experimental results including both simulation and real-world datasets look good. While this work emphasizes the communication complexity, it is better to see some results between accuracy and communication rounds, or something like that. Moreover, in this study, the algorithms are already well-established and the authors only revisit them to incorporate a new concept. Hence, the empirical evidence should cover sufficient benchmark datasets and models. Only including MNIST and CIFAR 10 with simple models overall makes the validation relatively weak. I suggest the authors should consider presenting more experimental results for the experimental validation.

---

### Official Review · Reviewer_eJky · 2022-10-16
**A new measure for heterogeneity in D-SGD**

This paper proposed a new measure for heterogeneity in D-SGD. Their set of assumptions 2-4 is more general than previous assumption 2,3,5. Based on their assumptions, a sharper bounded is derived.

Pros.

1. Their assumptions are more general and the derived bound is tighter.

2. A method of learning the topology is proposed to validate the assumption they proposed.

Cons.

1. The authors should provide a detailed comparison between their upper bound and other upper bounds for D-SGD to show the advantage. It is not clear if the advantage is only shown on the second term. Also, other terms are the same compared to others?

Q1, the only difference between results for convex and non-convex cases is $L$, does this mean there is no difference on the rate between this two?

Q2, It seems that Assumption is also related with $p$. Can this assumption be written as a form of $p$?

2. The application is limited to `Label Skew'.

Conclusion
Although there are limitations (only applied to Label Skew, no novel improvement for the convergence bound), and more needed to be done to consider the relationship between the new proposed assumption and $p$, this paper proposed a new measure of heterogeneity for D-SGD to substitute previous Assumption 5, which is worth thinking.

---

### Official Review · Reviewer_1eL7 · 2022-10-18
**new theory on D-SGD and some corrections based on the theory**

In this paper, the authors propose a newer analysis of the D-SGD algorithm with a new metric for a different perspective: "Data Heterogeneity." Further, the authors propose a modification of the W mixing matrix to minimize the problem caused by label skew. This will be a preprocessing step and will not change anything in the algorithm. The authors compare the proposed algorithm with different network topologies and show better performance. The results seems to show significant improvement over other compared methods. However, there are several other comparative methods other than D-Cliques for obtaining a sparse mixing matrix. The authors are recommended to compare against them too.

---

### Decision · Program_Chairs · 2022-10-20

Accept (Poster)